# Cooperative interactions enable singular olfactory receptor expression in mouse olfactory neurons

Kevin Monahan[1], Ira Schieren[1], Jonah Cheung[2], Alice Mumbey-Wafula[1], Edwin S Monuki[3], Stavros Lomvardas[1,4,5]*

[1]Department of Biochemistry and Molecular Biophysics, Columbia University, New York, United States; [2]New York Structural Biology Center, New York, United States; [3]Department of Pathology and Laboratory Medicine, School of Medicine, University of California Irvine, Irvine, United States; [4]Department of Neuroscience, Columbia University, New York, United States; [5]Zuckerman Mind Brain and Behavior Institute, Columbia University, New York, United States

**Abstract** The monogenic and monoallelic expression of only one out of >1000 mouse olfactory receptor (ORs) genes requires the formation of large heterochromatic chromatin domains that sequester the OR gene clusters. Within these domains, intergenic transcriptional enhancers evade heterochromatic silencing and converge into interchromosomal hubs that assemble over the transcriptionally active OR. The significance of this nuclear organization in OR choice remains elusive. Here, we show that transcription factors Lhx2 and Ebf specify OR enhancers by binding in a functionally cooperative fashion to stereotypically spaced motifs that defy heterochromatin. Specific displacement of Lhx2 and Ebf from OR enhancers resulted in pervasive, long-range, and *trans* downregulation of OR transcription, whereas pre-assembly of a multi-enhancer hub increased the frequency of OR choice in cis. Our data provide genetic support for the requirement and sufficiency of interchromosomal interactions in singular OR choice and generate general regulatory principles for stochastic, mutually exclusive gene expression programs.
DOI: https://doi.org/10.7554/eLife.28620.001

*For correspondence: sl682@columbia.edu

Competing interests: The authors declare that no competing interests exist.

## Introduction

The mammalian main olfactory epithelium (MOE) provides an extreme example of cellular diversity orchestrated by the seemingly stochastic, monogenic, and monoallelic expression of a single olfactory receptor (OR) gene. Each mature olfactory sensory neuron (mOSN) in the MOE expresses only one OR that is chosen from a pool of more than two thousand alleles (*Buck and Axel, 1991*; *Chess et al., 1994*). The basis of the regulation of OR gene expression is chromatin-mediated transcriptional silencing followed by the stochastic de-repression and, thereby, transcriptional activation of a single OR allele that prevents the de-repression of additional OR genes (*Dalton and Lomvardas, 2015*; *Monahan and Lomvardas, 2015*). OR gene clusters are assembled into constitutive heterochromatin at early stages of OSN differentiation (*Magklara et al., 2011*), a process that represses OR transcription and preserves the monogenic and stochastic nature of OR expression (*Lyons et al., 2014*). Heterochromatic silencing is reinforced by the interchromosomal convergence of OR loci to OSN-specific, highly compacted nuclear bodies that assure complete transcriptional silencing of ORs in mOSNs (*Clowney et al., 2012*). Consequently, OR gene activation requires de-silencing by lysine demethylase Lsd1 (*Lyons et al., 2013*) and spatial segregation of the single chosen OR allele towards euchromatic nuclear territories (*Armelin-Correa et al., 2014*; *Clowney et al., 2012*). Translation of the newly transcribed OR mRNA activates a co-opted arm of the unfolded

protein response (*Dalton et al., 2013*) and induces a feedback signal (*Lewcock and Reed, 2004*; *Serizawa et al., 2005*; *Shykind et al., 2004*) that turns off *Lsd1*, preventing the de-silencing and activation of additional OR genes (*Lyons et al., 2013*).

In the context of this repressive chromatin environment, OR gene choice requires the action of intergenic enhancers that escape heterochromatic silencing and activate the transcription of their proximal ORs (*Khan et al., 2011*; *Markenscoff-Papadimitriou et al., 2014*; *Serizawa et al., 2003*). These euchromatic enhancer 'islands', which we named after Greek Islands, engage in interchromosomal interactions with each other, and with the transcriptionally active OR allele, forming a multi-enhancer hub for OR transcription outside of the repressive OR foci (*Clowney et al., 2012*; *Lomvardas et al., 2006*; *Markenscoff-Papadimitriou et al., 2014*). The convergence of multiple Greek Islands to the chosen OR allele suggests that strong, feedback-eliciting OR gene transcription may be achieved only in the context of a multi-enhancer hub (*Markenscoff-Papadimitriou et al., 2014*). Yet, the molecular mechanisms that specify Greek Islands in the context of OR heterochromatin and, thus, enable their elaborate interactions during OSN differentiation remain unknown.

Here, we present a detailed molecular characterization of the Greek Islands, which revealed a common genetic signature and occupancy by shared sequence-specific transcription factors, allowing us, for the first time, to incapacitate them as a whole. ChIP-seq studies of FAC-sorted mOSNs revealed that most of the previously characterized Greek Islands, and several newly identified islands, are bound by two transcription factors: Lhx2 and Ebf. Computational analysis of the co-bound ChIP-seq peaks from Greek Islands revealed stereotypically positioned Lhx2 and Ebf binding sites that together constitute a 'composite' binding motif that affords cooperative binding in vivo. This motif is highly enriched in Greek Islands relative to OR promoters and Lhx2/Ebf co-bound sites genome-wide. Considering the prevalence and specificity of this composite motif in Greek Islands, we designed a synthetic 'fusion' protein that binds to this consensus sequence and not to individual Lhx2 or Ebf motifs in vitro. We found that overexpression of this fusion protein in mOSNs eliminated chromatin accessibility at most Greek Islands, and resulted in strong transcriptional downregulation of every OR, regardless of their genomic distance, or even their chromosomal linkage to a Greek Island. Finally, partial pre-assembly of a Greek Island hub in cis, by insertion of an array of 5 Greek Islands next to the Greek Island Rhodes, significantly increased the frequency of expression of Rhodes-linked OR genes. These manipulations provide genetic support for the requirement of *trans* enhancement in OR gene expression, and are consistent with the sufficiency of a multi-enhancer hub formation for OR gene choice.

## Results

### Greek Islands are co-bound by Lhx2 and EBF

Greek Islands share a characteristic chromatin modification signature and in vivo footprints for transcription factors Lhx2 and Ebf (*Markenscoff-Papadimitriou et al., 2014*). To test the predicted binding of Lhx2 and Ebf, we performed ChIP-seq experiments using crosslinked chromatin prepared from FAC-sorted mOSNs, the neuronal population that stably expresses ORs in a singular fashion. To isolate mOSNs we FAC-sorted GFP[+] cells from the MOEs of *Omp-IRES-GFP* knock-in mice, as previously described (*Magklara et al., 2011*). The Ebf antibody we used for these experiments cross-reacts with all 4 Ebf proteins, Ebf1-4, (data not shown), which are all highly expressed in the MOE. Because of the genetic redundancy of the Ebf genes in the MOE (*Wang et al., 2004*), and because the 4 Ebf members form homo- and hetero-dimers with identical sequence specificity (*Wang et al., 1997*), we did not attempt to further distinguish between the 4 paralogues. For Lhx2 ChIP-seq studies we used a custom-made antibody (*Roberson et al., 2001*). The specificity of these antibodies is supported by motif analysis of the Lhx2 and Ebf ChIP-seq experiments, which revealed that the Lhx2 and Ebf binding sites are the most highly enriched motifs respectively (*Figure 1A*). Genome-wide, we identified 9024 peaks for Ebf and 16,311 Lhx2 peaks, with 4792 peaks being co-bound by both proteins (*Figure 1B*). Despite the in vivo recognition of an essentially identical motif in pro/pre-B cells(*Györy et al., 2012*; *Kong et al., 2016*; *Treiber et al., 2010b*), where Ebf acts as master regulator of B-cell differentiation (*Mandel and Grosschedl, 2010*), there is little overlap between the genome-wide binding of Ebf in mOSNs and B-cell progenitors (data not shown). Genes proximal to Lhx2 and Ebf co-bound sites in mOSNs are statistically enriched for functions related to

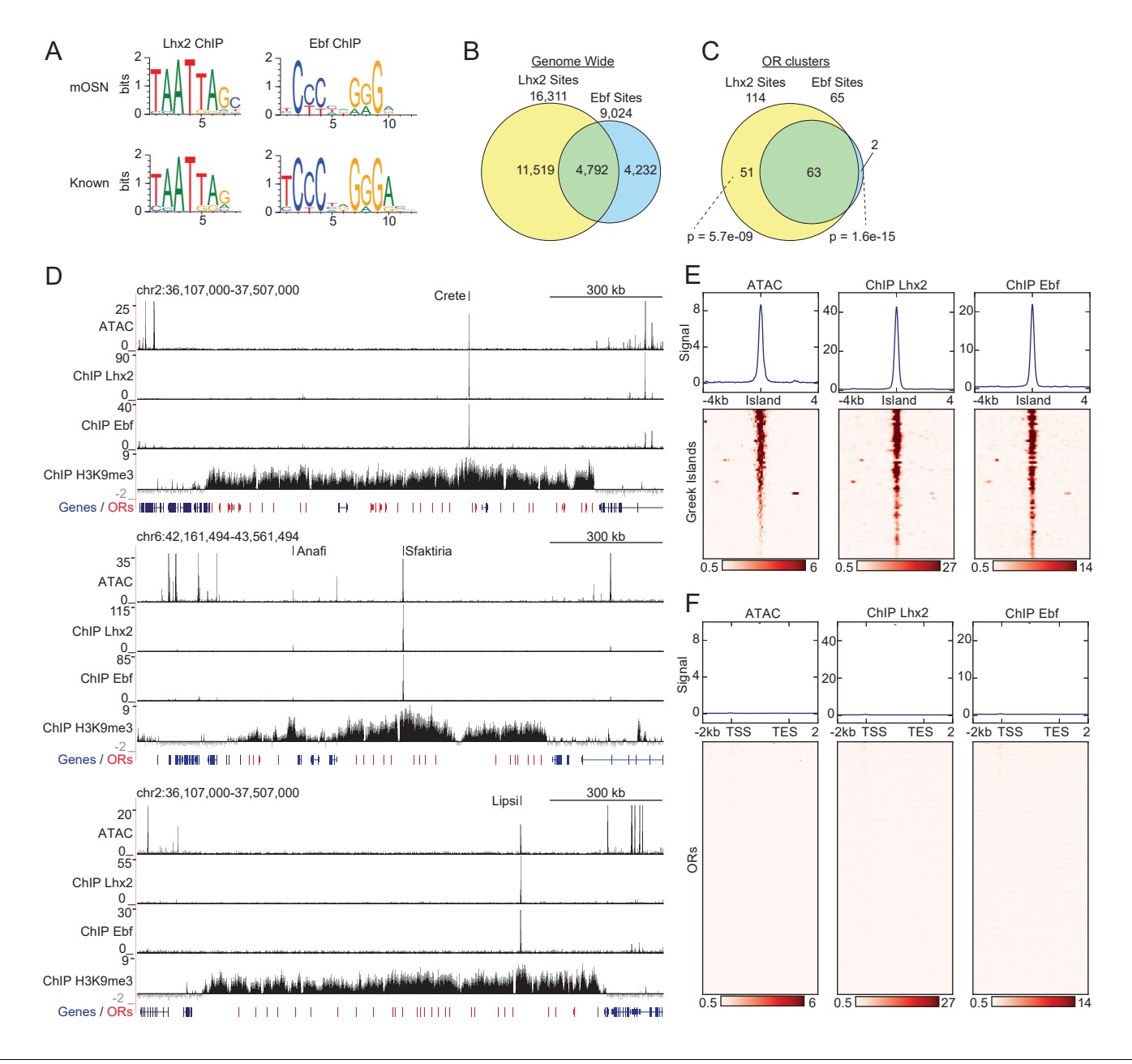

**Figure 1.** Greek Islands represent Lhx2 and Ebf co-bound regions residing in heterochromatic OR clusters. (A) The top sequence motif identified for mOSN ChIP-seq peaks is shown above sequence motifs generated from previously reported Lhx2 (*Folgueras et al., 2013*) and Ebf (*Lin et al., 2010*) ChIP-seq data sets. mOSN ChIP-seq peaks were identified using HOMER and motif analysis was run on peaks present in both biological replicates. (B) Overlap between mOSN Lhx2 and Ebf bound sites genome-wide. See *Figure 1—figure supplement 2* for analysis of ChIP-seq signal on Ebf and Lhx2 Co-bound sites within OR clusters. (C) Overlap between mOSN Lhx2 and Ebf bound sites within OR clusters. For each factor, co-bound sites are significantly more frequent within OR clusters than in the rest of the genome (p=5.702e$^{-9}$ for Lhx2, p=1.6e$^{-15}$ for Ebf, Binomial test). See *Figure 1—figure supplement 2* for gene ontology analysis of peaks bound by Lhx2 and Ebf. (D) mOSN ATAC-seq and ChIP-seq signal tracks for three representative OR gene clusters. Values are reads per 10 million. Below the signal tracks, OR genes are depicted in red and non-OR genes are depicted in blue. Greek Island locations are marked. *Anafi* is a newly identified Greek Island, located in a small OR cluster upstream of the *Sfaktiria* cluster. See also *Figure 1—figure supplement 3* and *Supplementary file 1*. For ATAC-seq, pooled data is shown from 4 biological replicates, for ChIP-seq, pooled data is shown from 2 biological replicates. For H3K9me3 ChIP-seq, input control signal is subtracted from ChIP signal prior to plotting. (E) mOSN ATAC-seq or ChIP-seq signal across 63 Greek Islands. Each row of the heatmap shows an 8 kb region centered on a Greek Island. Regions of high signal are shaded red. Mean signal across all elements is plotted above the heatmap, values are reads per 10 million. All heatmaps are sorted in the same order, based upon ATAC-seq signal. See also *Figure 1—figure supplement 3* and *Supplementary file 1*. For ATAC-seq, pooled

*Figure 1 continued on next page*

*Figure 1 continued*

data is shown from 4 biological replicates, for ChIP-seq, pooled data is shown from 2 biological replicates. See *Figure 1—figure supplement 4* for a comparison of newly and previously identified Greek Islands, and *Figure 1—figure supplement 5* for RNA-seq analysis of ORs with Greek Islands near the TSS. (F) mOSN ATAC-seq and ChIP-seq signal tracks on OR genes. Each row of the heatmap shows an OR gene scaled to 4 kb as well as the 2 kb regions upstream and downstream. Plots and heatmap are scaled the same as in *Figure 1E*.

DOI: https://doi.org/10.7554/eLife.28620.002

The following source data, source code and figure supplements are available for figure 7:

**Source code 1.** R code for analysis of ChIP-seq data from mOSNs.r.
DOI: https://doi.org/10.7554/eLife.28620.008
**Source code 2.** R code for analysis of RNA-seq data from mOSNs.r.
DOI: https://doi.org/10.7554/eLife.28620.009
**Source data 1.** Lhx2 ChIP-seq signal by peak type.txt.
DOI: https://doi.org/10.7554/eLife.28620.010
**Source data 2.** Ebf ChIP-seq signal by peak type.txt.
DOI: https://doi.org/10.7554/eLife.28620.011
**Source data 3.** Transcript level of ORs grouped by presence of Greek Island in Promoter.txt.
DOI: https://doi.org/10.7554/eLife.28620.012
**Figure supplement 1.** Gene Ontology terms associated with Ebf and Lhx2 co-bound sites.
DOI: https://doi.org/10.7554/eLife.28620.003
**Figure supplement 2.** Co-binding of Ebf and Lhx2 within OR clusters.
DOI: https://doi.org/10.7554/eLife.28620.004
**Figure supplement 3.** Histone modifications proximal to Greek Islands.
DOI: https://doi.org/10.7554/eLife.28620.005
**Figure supplement 4.** Comparison of new and previously identified Greek Islands.
DOI: https://doi.org/10.7554/eLife.28620.006
**Figure supplement 5.** Greek Islands represent Lhx2 and Ebf co-bound regions residing in heterochromatic OR clusters.
DOI: https://doi.org/10.7554/eLife.28620.007

olfactory transduction and axonogenesis (*Figure 1—figure supplement 1*), consistent with a combinatorial role of these transcription factors in OSN differentiation and function (*Hirota and Mombaerts, 2004*; *Wang et al., 1993*; *Wang et al., 2004*; *Wang et al., 1997*).

The apparent coordinated binding of Lhx2 and Ebf to genomic DNA is exaggerated within the boundaries of heterochromatic OR clusters where individually bound peaks are rare and have low signal. Specifically, there are 63 peaks that are co-bound by both Lhx2 and Ebf, 2 Ebf-only, and 51 Lhx2-only peaks (*Figure 1C*) in the ~36 MB of OR clusters, a significantly higher rate of overlap than the rate observed genome-wide ($p=1.5e^{-15}$ and $p=5.7e^{-9}$, respectively, Binomial test). Notably, most Ebf and Lhx2 co-bound sites in OR clusters have much stronger ChIP signal than singly bound sites (*Figure 1—figure supplement 2A*). Several of these co-bound sites within OR clusters are among the regions of highest ChIP-seq signal in the genome, suggesting that they are bound in a large fraction of mOSNs (*Figure 1—figure supplement 2B–C*), whereas individually bound peaks barely pass our peak-calling threshold. Co-bound sites within OR clusters coincide with 21 of the 35 previously characterized Greek Islands (*Supplementary file 1*). For example, visual inspection of three Greek Islands, *Crete*, *Sfaktiria* and *Lipsi*, revealed strong Lhx2 and Ebf binding despite the high levels of flanking H3K9me3 on these OR clusters (*Figure 1D*). ATAC-seq analysis in the same cellular population revealed increased chromatin accessibility at the exact genomic location of the Lhx2 and Ebf ChIP-seq peaks, but very little accessibility across the rest of the OR cluster (*Figure 1D*). Each of these sites also exhibits a reduction of the heterochromatic modifications, H3K9me3 and H3K79me3, over the body of the element, and locally increased levels of the active enhancer mark H3K7ac (*Figure 1—figure supplement 3A*). Overall, this chromatin signature is shared by the full set of Ebf and Lhx2 co-bound sites within OR gene clusters (*Figure 1E* and *Figure 1—figure supplement 3B*). Thus, Lhx2/Ebf co-bound sites that do not correspond to the original Greek Islands (*Supplementary file 1*) likely represent additional, less frequently active Islands that were only detected here due to the increased sensitivity of our mOSN-specific analysis (*Anafi* in *Figure 1D* and *Figure 1—figure supplement 4* for comparison between old and new Islands). In contrast, Greek Islands from the original set that lack Ebf and Lhx2 binding in mOSNs also deviate

from the characteristic 'epigenetic' signature obtained from whole MOE experiments (*Supplementary file 1*). Thus, these sites are likely to be functionally distinct or active in a different population of cells within the MOE, and are not included within our revised set of Greek Islands.

OR gene promoters are also significantly enriched for predicted Lhx2 and Ebf binding sites (*Clowney et al., 2011*; *Michaloski et al., 2006*; *Plessy et al., 2012*; *Young et al., 2011*), and mutations of individual Ebf and Lhx2 sites have been shown to reduce OR expression in vivo (*Rothman et al., 2005*). However, as a whole, OR gene promoters are inaccessible and not bound by these transcription factors in mOSNs (*Figure 1F*). Specifically, only 10 OR promoters show significant binding of Ebf and Lhx2 within 500 bp of the TSS. Interestingly, these 10 ORs are expressed at levels similar to the median of OR expression (*Figure 1—figure supplement 5* and *Supplementary file 1*). Thus, detection of Lhx2 and Ebf binding on these peaks is not explained by the unusually frequent transcriptional activation of their proximal ORs.

## OR identity does not affect Greek Island accessibility

Based on the observation that most OR promoters display a complete lack of chromatin accessibility and Lhx2/Ebf binding, we asked if these promoters are accessible to transcription factors only in the OSNs that transcribe them. We FAC-sorted OSNs that express the same OR allele, by isolating GFP$^+$ cells from Olfr17-IRES-GFP (*Gogos et al., 2000*), Olfr151-IRES-tauGFP (*Bozza et al., 2002*), and Olfr1507-IRES-GFP (*Shykind et al., 2004*) knock-in mice (*Figure 2A,B*), and performed ATAC-seq (*Buenrostro et al., 2013*). As expected, the promoters Olfr1507, Olfr17 and Olfr151, are highly accessible when these genes are transcriptionally active (*Figure 2C*), consistent with local chromatin de-compaction being a prerequisite for OR gene transcription (*Magklara et al., 2011*). We also detect an increase in transposase accessibility at the 3'UTR of transcriptionally active OR alleles, an unusual feature that is not characteristic of most transcriptionally active genes in OSNs (*Figure 2C*, *Figure 2—figure supplement 1*).

In contrast to the differences between active and silent OR promoters, the overall pattern of accessibility of the Greek Islands is very similar in OSN populations that have chosen different ORs (*Figure 2D*). Very few Greek Islands display significantly different accessibility in the three OSN populations when compared to mOSNs (*Figure 2E*), and most fluctuations represent small but uniform shifts in Greek Island accessibility. For example, the H enhancer, which is proximal to Olfr1507 and is required for Olfr1507 expression, has a relatively strong ATAC-seq signal in all four cell populations and is not significantly stronger in Olfr1507+ cells than in mOSNs (*Figure 2D,E*, *Figure 2—figure supplement 2A*). However, we do note some evidence for differential activity of Greek Islands. In particular, *Kimolos*, the Greek Island proximal to Olfr151, has relatively weak ATAC-seq signal in mOSN and in Olfr17+ and Olfr1507+ OSNs, but exhibits a nearly 10-fold increase in signal in Olfr151-expressing cells (*Figure 2D,E*, *Figure 2—figure supplement 2B*). Thus, it appears that a large number of Greek Islands are broadly accessible in most OSNs, irrespective of the identity of the chose OR allele, whereas OR promoters are accessible only in the OSNs in which they are active.

## Proximity of Lhx2/Ebf motifs correlates with binding on Greek Islands

What mechanism allows binding of Lhx2 and Ebf on Greek Islands but not OR promoters in most OSNs? We hypothesized that additional factors may bind specifically on Greek Islands but not on OR promoters, providing the functional distinction between the two types of regulatory elements. Motif analysis of the Lhx2 and Ebf ChIP-seq peaks using HOMER (*Heinz et al., 2010*) did not reveal additional known DNA binding sites that are shared by a significant portion of Greek Islands, other than Lhx2 and Ebf. De novo motif analysis, however, uncovered a novel, 'composite' motif that corresponds to Lhx2 and Ebf sites positioned next to each other (*Figure 3A*). This composite Lhx2/Ebf motif is structurally very similar to the numerous heterodimeric motifs identified by an in vitro screen for sequences that are co-bound by a variety of transcription factor combinations (*Jolma et al., 2015*). A stringent Lhx2/Ebf composite motif, with score over 10 (see material and methods), is found in 35 of the 63 Greek Islands (*Figure 3—figure supplement 1A*, *Supplementary file 2*). This motif is significantly enriched in Greek Islands in comparison with OR promoters and with Lhx2/Ebf co-bound peaks outside of OR clusters (*Figure 3B*). In aggregate, the 43 strong composite motifs found in Greek Islands reside exactly at a local depletion of the ATAC-seq signal from mOSNs,

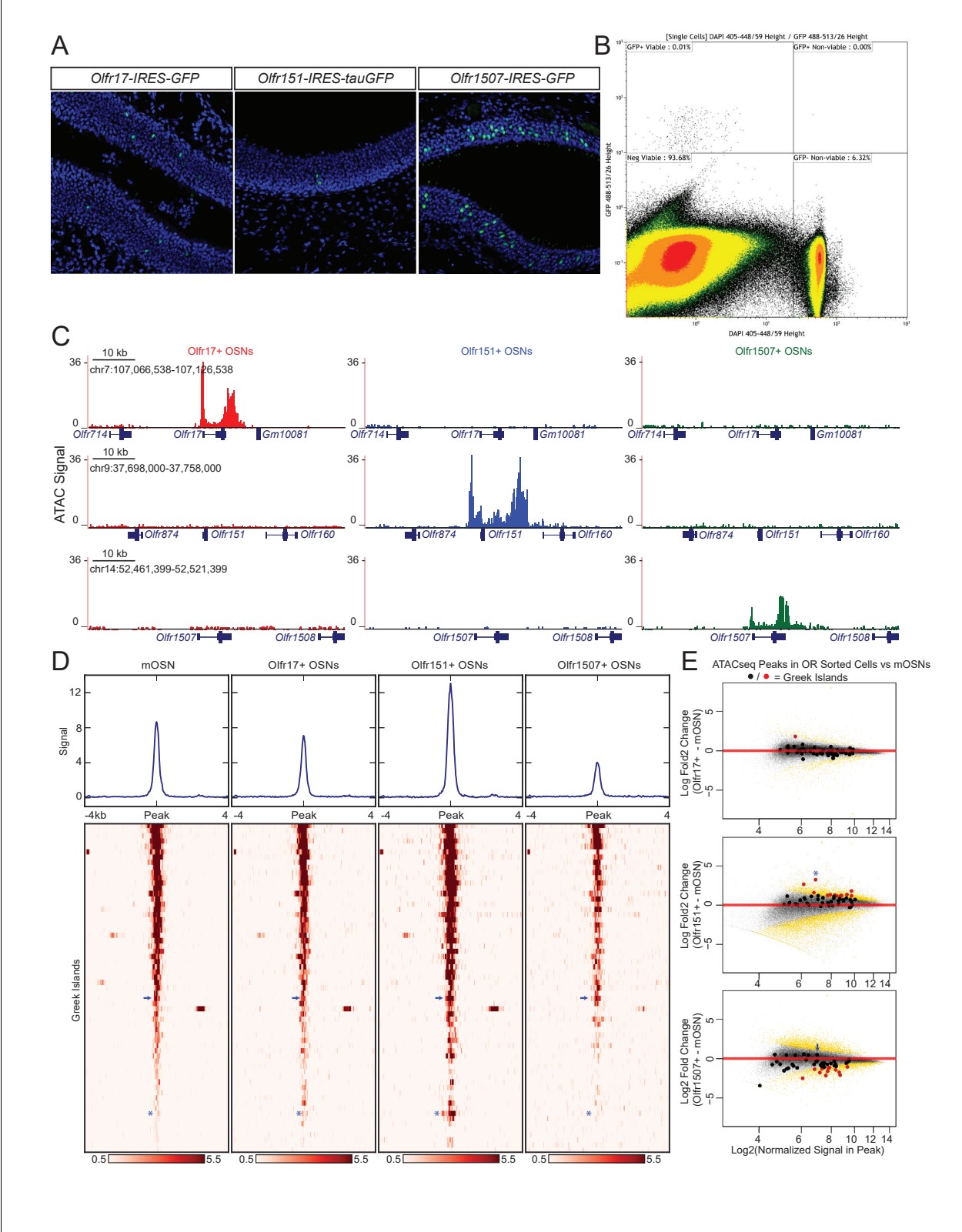

**Figure 2.** Greek island accessibility is independent of OR promoter choice. (**A**) GFP fluorescence (green) in MOE tissue sections from adult mice bearing *Olfr17-IRES-GFP*, *Olfr151-IRES-tauGFP*, or *Olfr1507-IRES-GFP* alleles. Nuclei are stained with DAPI (blue). (**B**) Representative FACS data for *Olfr-IRES-GFP* mice. Data is shown from *Olfr151-IRES-GFP* mice. Viable (DAPI negative), GFP+ cells were collected for ATAC-seq. (**C**) ATAC-seq signal tracks from GFP+ cells sorted from *Olfr17-IRES-GFP* (red), *Olfr151-IRES-GFP* (blue), or *Olfr1507-IRES-GFP* (green) mice. Values are reads per 10 million.
*Figure 2 continued on next page*

*Figure 2 continued*

The region spanning each targeted OR is shown for all three lines. See also *Figure 2—figure supplement 1*. Pooled data is shown for 2 biological replicates. (D) ATAC-seq signal over Greek Islands is shown for mOSNs and each *Olfr-IRES-GFP* line. All samples are sorted by signal in mOSNs. A blue arrow marks the H Enhancer, which is the Greek Island proximal to *Olfr1507*. A blue asterisk marks *Kimolos*, the Greek Island proximal to *Olfr151*, which has the strongest change in signal relative to mOSNs. See also *Figure 2—figure supplement 2*. Pooled data is shown for 4 biological replicates for mOSNs, and 2 biological replicates for each *Olfr-IRES-GFP* sorted population. (E) MA-plots showing fold change in ATAC-seq signal for each sorted *Olfr-IRES-GFP* population compared to mOSNs. Peak strength (normalized reads in peak) and fold change are shown for all ATAC-seq peaks; peaks that are not significantly changed are black and peaks that are significantly changed (FDR < 0.001) are gold. Greek Islands are plotted as larger dots and are shown in red if significantly changed. *Kimolos* is marked with an asterisk in Olfr151 expressing cells, and H is marked with an arrow in Olfr1507 expressing cells. See also *Figure 2—figure supplement 2*.

DOI: https://doi.org/10.7554/eLife.28620.013

The following source data, source code and figure supplements are available for figure 7:

**Source code 1.** R code for analysis of ATAC-seq data from OR-IRES-GFP.r.
DOI: https://doi.org/10.7554/eLife.28620.016
**Source data 1.** ATAC-seq MA plot of mOSN versus Olfr17-ires-GFP.txt.
DOI: https://doi.org/10.7554/eLife.28620.017
**Source data 2.** ATAC-seq MA plot of mOSN versus Olfr151-ires-tauGFP.txt.
DOI: https://doi.org/10.7554/eLife.28620.018
**Source data 3.** ATAC-seq MA plot of mOSN versus Olfr1507-ires-GFP.txt.
DOI: https://doi.org/10.7554/eLife.28620.019
**Figure supplement 1.** High Accessibility near the OR transcription end site.
DOI: https://doi.org/10.7554/eLife.28620.014
**Figure supplement 2.** Greek island accessibility is independent of OR promoter choice.
DOI: https://doi.org/10.7554/eLife.28620.015

consistent with in vivo occupancy of these sequences by transcription factors (*Figure 3C*) as previously described (*Buenrostro et al., 2015*).

Visual inspection of the aligned composite motifs revealed that the Ebf site is less constrained to stretches of C and G bases than solitary Ebf motifs, and instead tolerates stretches of pyrimidines and purines that retain a highly stereotypic spacing from the Lxh2 site (*Figure 3D,E*, top panel). Recent observations suggested that the relative positioning of DNA binding motifs compensates for the fluctuation of individual nucleotides in vivo (*Farley et al., 2016*). Similarly, the positioning of transcription factors on the face of the DNA double helix, as determined by the spacing between transcription factor binding sites, is more important than the relative strength of individual binding sites for the assembly of the *IFN beta* enhanceosome (*Merika et al., 1998*; *Thanos and Maniatis, 1995*). Thus, we asked if composite motifs with lower scores, which, predominantly, have degenerate Ebf motifs (*Figure 3—figure supplement 1B*), still meet these stereotypic constraints. Indeed, despite increased fluctuation in the nucleotide level, the stereotypic distribution between purines and pyrimidines is retained in composites with score above 5 (*Figure 3D,E* bottom panel), with a new total of 55 out of 63 Greek Islands having a composite motif under this less stringent cutoff. Moreover, of the 28 Greek Islands that lack a strong composite, 20 have an Ebf site that is juxtaposed to an Lhx2 site. The distance between Ebf and Lhx2 sites in these Greek Islands is significantly shorter than the distance between Ebf and Lhx2 sites in OR promoters and in co-bound peaks outside of OR gene clusters (*Figure 3F*). In total, 61/63 islands contain a composite motif and/or very proximal Lhx2 and Ebf binding sites (*Supplementary file 2*). Thus, although Lhx2 and Ebf frequently bind at the same genomic targets genome-wide, their binding on Greek Islands is restricted to stereotypically proximal Lhx2 and Ebf motifs.

## Lhx2 is essential for Ebf binding on Greek Islands

An immediate prediction of our computational analyses is that Lhx2 and Ebf bind cooperatively to composite DNA binding motifs. In addition, Lhx2 and Ebf binding to these stereotypically spaced motifs may result in synergistic recruitment of coactivators that cannot be recruited by the individually bound proteins. In either case of functional cooperativity, deletion of either Lhx2 or Ebf should abolish the binding of the other transcription factor on Greek Islands. To test this we deleted *Lhx2* from mOSNs, using a conditional *Lhx2* allele (*Mangale et al., 2008*) that we crossed to *Omp-IRES-*

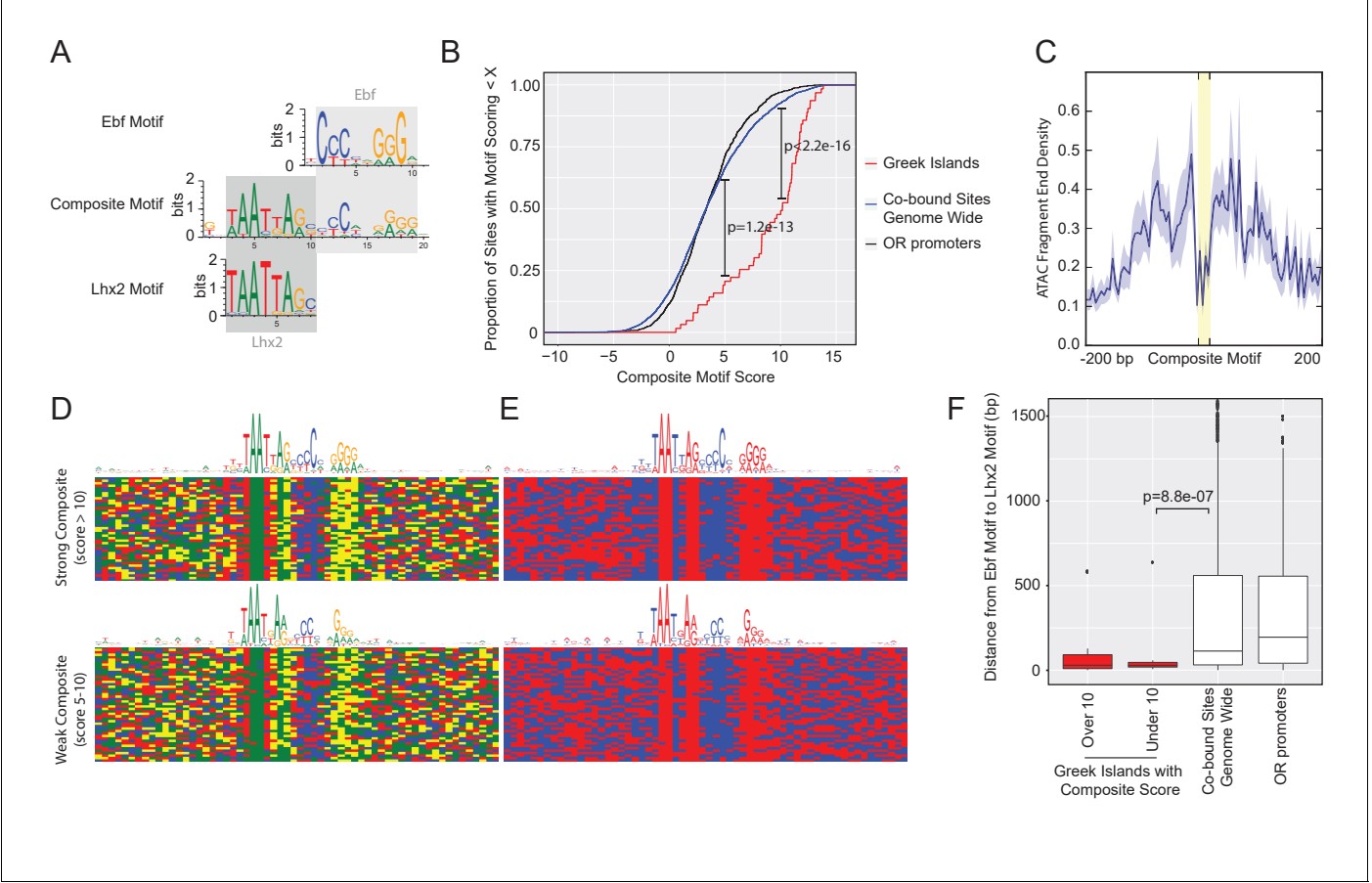

**Figure 3.** Greek Islands have stereotypically proximal Lhx2 and Ebf motifs. (**A**) Sequence logo of the Greek Island composite motif (center). The mOSN ChIP-seq derived Lhx2 and Ebf motifs logos are positioned above and below the corresponding regions of the composite motif. (**B**) Cumulative distribution plot of the score of the best composite motif site found in each of the 63 Greek Islands. Also plotted are cumulative distributions for co-bound sites outside of OR clusters and OR gene promoters. A score of 10 was selected as a stringent threshold for motif identification, and a score of 5 was selected for permissive motif identification. This motif is significantly enriched in Greek Islands relative to co-bound sites outside of OR clusters at both of these score cut-offs (Binomial test). See also *Supplementary file 2*. (**C**) Plot of the density of ATAC-seq fragment ends in the vicinity of Greek Island composite motifs sites scoring over 10. Plot shows mean signal and standard error in 5 bp windows centered on 43 composite motif sites (yellow). (**D**) Multiple alignment of composite motif sequences from Greek Islands together with 20 bp of flanking sequence. Each base is shaded by nucleotide identity: A = green, C = blue, G = yellow, T = red. Top panel depicts composite with score over 10 and bottom panel depicts composites with score between 5 and 10, together with a sequence logo of the motif present in those sequences. See *Figure 3—figure supplement 1* for sequences of strong and weak Greek Island composite motifs. (**E**) As in (**D**), except purines are shaded red and pyrimidines are shaded blue. (**F**) For each site, the distance (in base pairs) between the closest Ebf-Lhx2 motif pair was determined. For each set of sites, the distribution of distances is shown as a boxplot. Sets of sites comprising Greek Islands with a strong composite motif, Greek Islands without a strong composite motif, Ebf and Lhx2 co-bound sites genome-wide, and OR gene promoters are compared. Sites without an Ebf motif are excluded. The distribution of distances between Ebf and Lhx2 motifs was significantly smaller for Greek Islands without a composite motif than for Ebf and Lhx2 bound sites genome-wide (two-sample, one-sided Kolmogorov–Smirnov test) See also *Supplementary file 2*. n = 25 for Greek Islands with Composite Score greater than 10; n = 21 for Greek Islands with Composite Score less than 10; n = 3805 for Co-bound sites genome wide; n = 521 for OR promoters.

DOI: https://doi.org/10.7554/eLife.28620.020

The following source data, source code and figure supplements are available for figure 7:

**Source code 1.** R code for Motif Analysis.r.
DOI: https://doi.org/10.7554/eLife.28620.022
**Source data 1.** Composite Motif Score Cumulative Distribution.txt.
DOI: https://doi.org/10.7554/eLife.28620.023
**Source data 2.** Motif Proximity.txt.
DOI: https://doi.org/10.7554/eLife.28620.024
**Figure supplement 1.** Greek Islands have stereotypically proximal Lhx2 and Ebf motifs.
DOI: https://doi.org/10.7554/eLife.28620.021

*Cre* mice. Deletion of *Lhx2* with *Omp-IRES-Cre*, results in loss of Lhx2 immunofluorescence (IF) signal from mOSNs, while Lhx2 protein levels are unaffected in progenitor and immature OSNs (*Figure 4A*). To enrich for *Lhx2* KO mOSNs in our analyses, we introduced the Cre-inducible fluorescent reporter tdTomato (*Madisen et al., 2010*) to our genetic strategy and we FAC-sorted Tomato$^+$ *Lhx2*-/- mOSNs. RNA-seq of the FAC-sorted cells verifies the deletion of the floxed exons in mOSNs and the generation of a mutant *Lhx2* mRNA that does not encode for Lhx2 protein (*Figure 4—figure supplement 1*). *Lhx2* gene deletion results in significant downregulation of OR gene expression (*Figure 4B*), a result consistent with the partial deletion of a different floxed *Lhx2* allele from mOSNs (*Zhang et al., 2016*). Furthermore, upon *Lhx2* deletion the Lhx2 ChIP-seq signal is depleted genome-wide and from the Greek Islands (*Figure 4C,D*). Importantly, deletion of *Lhx2* in mOSNs, results in loss of Ebf binding from Lipsi (*Figure 4C*) and from nearly all other Greek Islands (*Figure 4E*). ATAC-seq on the *Lhx2* KO OSNs also shows strong reduction of ATAC-peaks from Greek Islands (*Figure 4F*), suggesting that Lhx2 and Ebf co-binding on Greek Islands is essential for their sustained accessibility in this heterochromatic environment. Consistent with the role of composite motifs on cooperative Lhx2 and Ebf binding, the effects of *Lhx2* deletion on Ebf binding are weaker at co-bound sites outside the OR clusters compared to Greek Islands (*Figure 4G*, *Figure 4—figure supplement 2*). Interestingly, the general downregulation of OR gene transcription upon *Lhx2* deletion extends to ORs that do not have Lhx2 motifs on their promoters (*Figure 4—figure supplement 3*), suggesting that Lhx2 activates OR transcription predominantly through the Greek Islands.

## Inhibition of Greek Islands inhibits OR transcription

Our data suggest that composite motifs are an ideal target for genetic manipulations that could inhibit the function of Greek Islands as a whole. We reasoned that if we could fuse Lhx2 and Ebf DNA binding domains (DBD) at a proper distance, we could generate a DNA binding peptide that has high affinity for the composite but not for individual motifs. Because the DNA binding specificity of homeobox genes is low and is influenced by their partners (*Chan et al., 1994*; *Passner et al., 1999*), the Lhx2 DBD could be easily incorporated in this design. Ebf, however, has high affinity and specificity for its cognate palindromic motif, where it binds as a dimer (*Hagman et al., 1993*; *Hagman et al., 1995*; *Travis et al., 1993*; *Wang and Reed, 1993*; *Wang et al., 1997*). Crystal structure of an Ebf1 homodimer bound to DNA revealed that each DBD monomer contacts both halves of the palindromic motif and forms a clamp-like structure that likely stabilizes DNA binding (*Treiber et al., 2010a*). Thus, in order to reduce Ebf affinity for DNA without affecting its sequence specificity, we fused only one Ebf DBD to the Lhx2 DBD with various flexible linkers. Fusion of the two DNA binding domains with a 20aa protein linker generated a protein with affinity for the composite motif but not for individual Lhx2 and Ebf sites in vitro (*Figure 5A*). Competition experiments demonstrate that only unlabeled oligos containing the composite, and not individual Lhx2 or Ebf motifs, can compete off the binding of the fusion protein to the composite motif at up to 100x molar excess (*Figure 5B,C*). Remarkably, insertion of only 2 DNA bases between the Lhx2 and the Ebf binding sites on the composite motif impairs its ability to compete with the wild type composite (*Figure 5C*). Further increase of the distance between the two sites essentially eliminates any competitive advantage the composite motif had over the individual Lhx2 and Ebf sites (*Figure 5C*). Thus, the fusion of the Lhx2 DBD to a single Ebf DBD creates a novel DNA binding protein that recognizes the composite motif with sensitivity to the stereotypical distance of the two individual DNA binding sites.

To express the fusion protein in the MOE, we generated a transgenic construct under the control of the tetO promoter. This transgene includes a bi-cistronic mCherry reporter using the 2A peptide (*Kim et al., 2011*) (*Figure 5—figure supplement 1A*), which allows isolation of the transgene-expressing OSNs by FACS. We analyzed two independent founders, which we crossed to *Omp-IRES-tTA* knock-in mice (*Gogos et al., 2000*), to obtain expression of the fusion protein specifically in mOSNs (*Figure 5—figure supplement 1B*). We hypothesized that the fusion protein will compete with endogenous Lhx2 and Ebf for binding on composite motifs, acting as a repressor of the Greek Islands (*Figure 5D*). Indeed, ATAC-seq analysis shows strong reduction of ATAC-seq signal from the Greek Islands upon expression of the fusion protein in mOSNs (*Figure 5E,F*), suggesting the displacement of the heterochromatin-resisting transcription factors from OR enhancers. Unfortunately, both the Lhx2 and the Ebf antibodies we used in our ChIP-seq experiments cross-react with the DBD

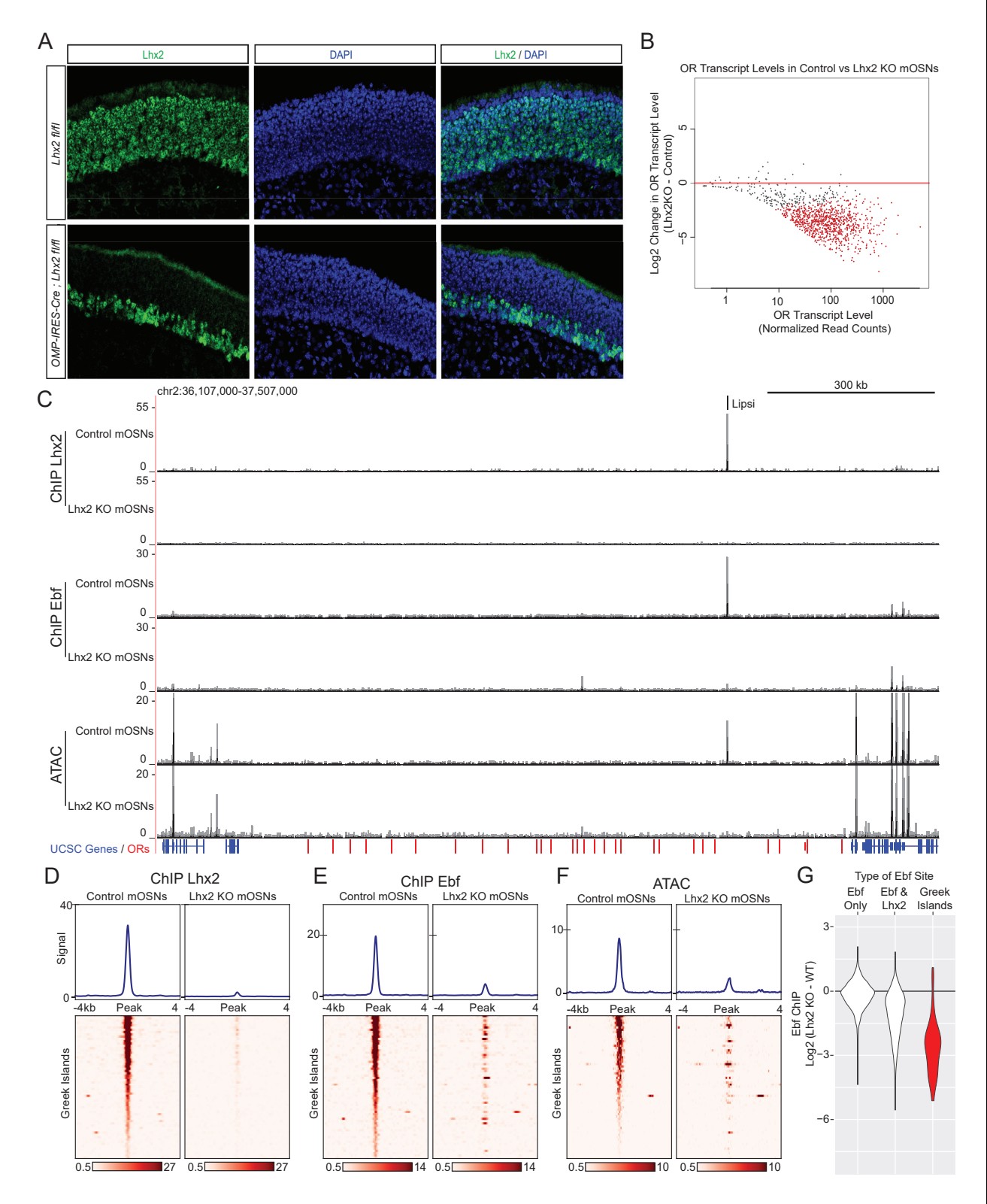

**Figure 4.** Lhx2 is required for Ebf binding predominantly on Greek Islands. (**A**) Lhx2 immunofluorescence (IF) (green) in MOE sections from 3 week old control (*Lhx2* fl/fl) and *Lhx2* KO (*Omp-IRES-Cre; Lhx2*fl/fl) mice. Nuclei are stained with DAPI (blue). The Lhx2 immunoreactive cells on the basal layers of the MOE represent immature OSNs and progenitors that have not yet turned on OMP (and thus Cre) expression. See also *Figure 4—figure supplement 1* for demonstration of the Cre induced deletion at the mRNA level. (**B**) MA-plot of OR transcript levels in FAC-sorted *Lhx2* KO mOSNs

*Figure 4 continued*

(*Omp-IRES-Cre; Lhx2fl/fl; tdTomato*) compared to FAC-sorted control mOSNs (*Omp-IRES-GFP*). Red dots correspond to OR genes with statistically significant transcriptional changes (adjusted p-value<0.05). Three biological replicates were included for control mOSNs and 2 biological replicates were included for *Lhx2* KO mOSNs. (C) ChIP-seq and ATAC-seq signal tracks from FAC-sorted control mOSNs (*Omp-IRES-GFP*) and *Lhx2* KO mOSNs (*Omp-IRES-Cre; Lhx2fl/fl; tdTomato*) for the OR cluster containing the Greek Island Lipsi. Values are reads per 10 million. For ATAC-seq, pooled data from 4 biological replicates for control mOSNs are compared to data from 2 biological replicates for *Lhx2* KO mOSNs. For ChIP, pooled data is shown from 2 biological replicates. (D–F) Heatmaps depicting Lhx2 and Ebf ChIP-seq and ATAC-seq signal across Greek Islands for FAC-sorted control and *Lhx2* KO mOSNs for the samples described in C. (G) Log2 fold change in normalized Ebf ChIP-seq signal in *Lhx2* KO mOSNs relative to control mOSNs for Greek Islands (red), compared to sites genome-wide that are bound by Ebf-only or both Ebf and Lhx2 in wild-type mOSNs. Fold change was calculated using data from 2 biological replicates each of control mOSNs and *Lhx2* KO mOSNs.. See also *Figure 4—figure supplement 2* for MA-plot showing data for all peaks in each set and *Figure 4—figure supplement 3* for RNA-seq analysis of the effect of Lhx2 KO on ORs with and without a promoter Lhx2 motif.

DOI: https://doi.org/10.7554/eLife.28620.025

The following source data, source code and figure supplements are available for figure 7:

**Source code 1.** R Code for analysis of ChIP-seq data from Lhx2KO mOSNs.r.
DOI: https://doi.org/10.7554/eLife.28620.029

**Source code 2.** R code for analysis of RNA-seq data from Lhx2KO mOSNs.r.
DOI: https://doi.org/10.7554/eLife.28620.030

**Source data 1.** RNA-seq MA plot of Olfr Expression in mOSNs versus Lhx2KO.txt.
DOI: https://doi.org/10.7554/eLife.28620.031

**Source data 2.** Effect of Lhx2KO on Ebf ChIPSeq signal.txt.
DOI: https://doi.org/10.7554/eLife.28620.032

**Source data 3.** MA-plot of Ebf ChIP-seq in control mOSNs versus Lhx2KO mOSNs.txt.
DOI: https://doi.org/10.7554/eLife.28620.033

**Source data 4.** Change in OR expression in Lhx2KO mOSNs versus promoter motifs.txt.
DOI: https://doi.org/10.7554/eLife.28620.034

**Figure supplement 1.** Effect of Lhx2 deletion on Lhx2 expression and splicing.
DOI: https://doi.org/10.7554/eLife.28620.026

**Figure supplement 2.** Lhx2 is required for Ebf binding predominantly on Greek Islands.
DOI: https://doi.org/10.7554/eLife.28620.027

**Figure supplement 3.** Lhx2 deletion downregulates ORs that do not have Lhx2 promoter motfis.
DOI: https://doi.org/10.7554/eLife.28620.028

domains of the fusion protein (data not shown), thus we could not confirm by ChIP-seq their displacement from the Greek Islands. However, RNA-seq analysis of the FAC-sorted mCherry+ cells revealed significant reduction of OR transcription as a whole (*Figure 5E,G*). Although the repressing effect of the fusion protein does not extend to non-OR genes residing outside of OR clusters (*Figure 5E*), it has a ubiquitous repressive effect on OR transcription (*Figure 5G*). In fact, of the 500 most significantly downregulated genes 482 are ORs (p<1e-313, hypergeometric test). In agreement with this, genome-wide analysis shows that while ORs are homogeneously repressed by the fusion protein, genes containing Ebf-, Lhx2-, or Ebf and Lhx2-bound promoters are, on average, transcriptionally unaffected (*Figure 5H,I*). Consistently, the effects of fusion protein overexpression on the ATAC signal of promoters bound by Lhx2 and/or Ebf are much weaker than the effects on Greek Islands (*Figure 5—figure supplement 2*). Finally, it is worth noting that similarly to the effects of the Lhx2 deletion, the repressive effects of the fusion protein on OR transcription does not depend on the presence of Ebf and Lhx2 motifs on OR promoters (*Figure 5—figure supplement 3*), supporting the Greek Island-mediated repressive effects of the fusion protein.

## Multi-enhancer hubs activate OR transcription

The widespread downregulation of OR gene expression detected in *Lhx2* KO and fusion protein expressing mOSNs suggests that the effects of Greek Island inhibition extend over large genomic distances, or even across chromosomes. Visual inspection of an isolated OR cluster on chromosome 16, which does not contain a Greek Island and is over 15 MB away from the closest OR cluster with a Greek Island, supports the strong downregulation of ORs in trans (*Figure 6—figure supplement 1A*). Genome-wide, for both *Lhx2* KO and fusion protein expressing mOSNs, there is a uniform reduction in OR expression regardless of the presence of a Greek Island in a cluster (*Figure 6A,B*).

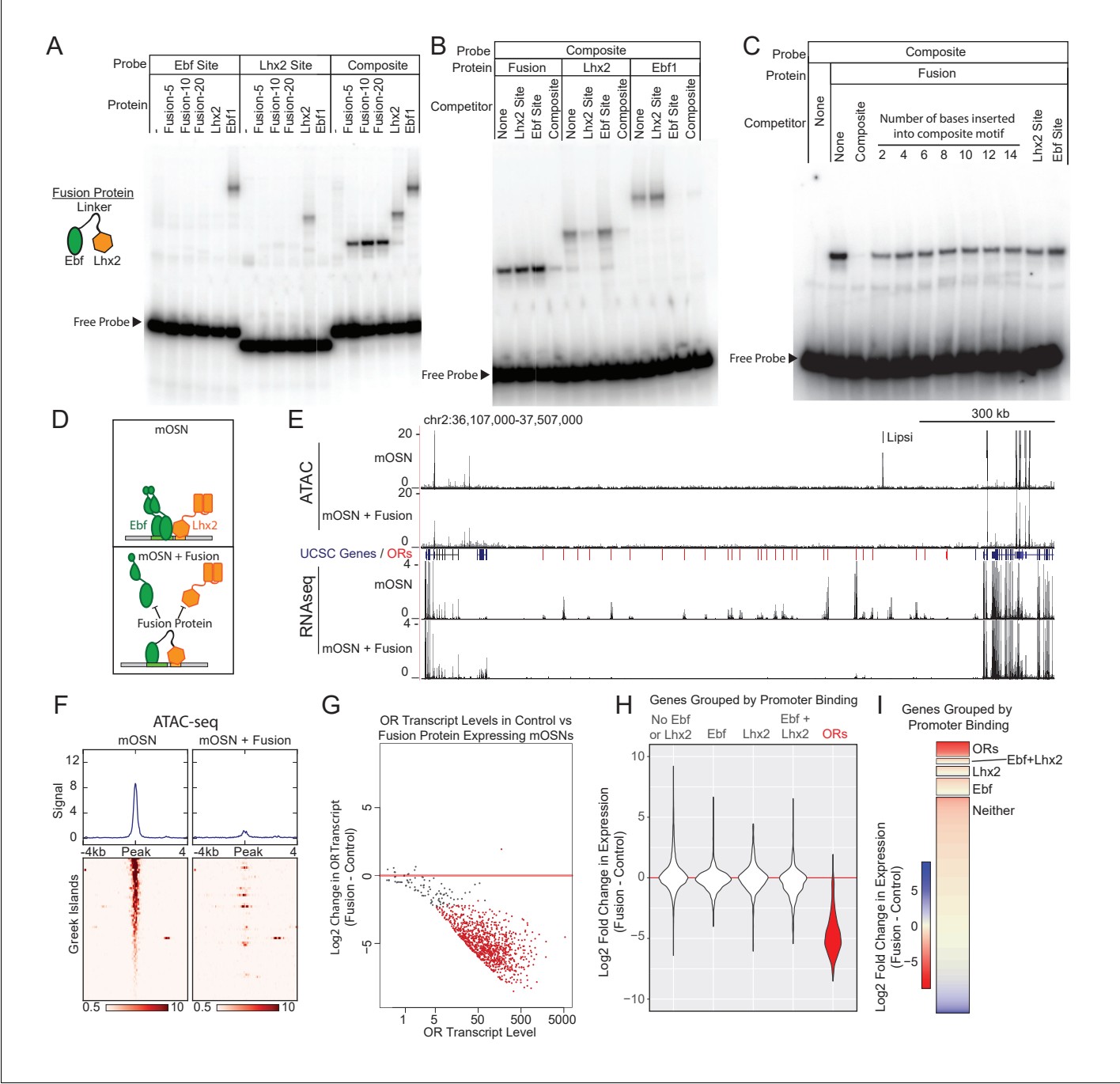

**Figure 5.** Displacement of Lhx2 and Ebf from Greek Islands shuts off OR transcription. (**A**) Electrophoretic Mobility Shift Assay (EMSA) for binding of in vitro translated protein to DNA probes containing either an Ebf site, an Lhx2 site, or a composite site. Binding of three versions of the Fusion protein with either 5, 10, or 20 amino acid linker peptides were compared to full length Lhx2 or full length Ebf1. (**B**) EMSA for sequence selectivity of in vitro translated proteins. Binding of Fusion protein (20aa linker), Ebf1, and Lhx2 to composite motif probe was competed with a 20-fold molar excess of unlabeled oligo containing either an Lhx2 site, Ebf site, or composite site. (**C**) EMSA for motif-spacing selectivity of in vitro translated proteins. Binding of Fusion protein (20aa linker) was competed with 100-fold molar excess of unlabeled oligo containing either wild type composite sequence or mutant composite generated by the insertion of 2–14 base pairs in two base pair increments. In the last two lanes the competitors are either a single Lhx2 or a single Ebf site. (**D**) Schematic illustrating the proposed dominant-negative activity of the fusion protein for composite motif sites. See also ***Figure 5—figure supplement 1*** for depiction of the genetic strategy for mOSN overexpression. (**E**) ATAC-seq and RNA-seq signal tracks from FAC-sorted control mOSNs and Fusion protein-expressing mOSNs for the OR cluster containing the Greek Island Lipsi. ATAC-seq values are reads per 10 million. RNA-seq values are reads per million. For ATAC-seq, pooled data from 4 biological replicates for control mOSNs are compared to data pooled from 2 independent founders of the Fusion Protein transgene. For RNA-seq, representative tracks are shown for one of three biological replicates for control

*Figure 5 continued on next page*

*Figure 5 continued*

mOSNs and for one of 2 independent founders for the Fusion Protein transgene. (**F**) ATAC-seq signal across the Greek Islands for control mOSNs and Fusion protein-expressing mOSNs. Pooled data from 4 biological replicates for control mOSNs are compared to data pooled from 2 independent founders of the Fusion Protein transgene. See *Figure 5—figure supplement 2* for the effect of Fusion Protein expression on Ebf and Lhx2 sites genome-wide. (**G**) MA-plot (*Dudoit and Fridlyand, 2002*) of OR transcript levels in FAC-sorted mOSNs expressing fusion protein (*Omp-IRES-tTA; tetO-Fusion-2a-mcherry*) compared to FAC-sorted control mOSNs (*Omp-IRES-GFP*). Red dots correspond to OR genes with statistical significant transcriptional changes (adjusted p-value<0.05). Three biological replicates were included for control mOSNs and data from 2 independent founders were included for the Fusion Protein transgene. See *Figure 5—figure supplement 3* for analysis of effect of Fusion Protein expression on ORs grouped by the presence of Ebf and Lhx2 promoter motifs. (**H**) Violin plot of Log2 fold change in transcript levels of ORs (red) in mOSNs expressing fusion protein compared to control mOSN. ORs are compared to additional sets of genes: genes with Ebf and Lhx2 bound within 1 kb of the TSS, genes with Lhx2-only bound within 1 kb of the TSS, genes with Ebf-only bound within 1 kb of the TSS, and non-OR genes without Ebf or Lhx2 binding. (**I**) As in (**H**), with Log2 fold change in transcript levels shown as a heatmap for each set of genes.
DOI: https://doi.org/10.7554/eLife.28620.035

The following source data, source code and figure supplements are available for figure 7:

**Source code 1.** R code for analysis of ATAC-seq data from Fusion Protein mOSNs.r.
DOI: https://doi.org/10.7554/eLife.28620.039
**Source code 2.** R code for analysis of RNA-seq data from Fusion Protein mOSNs.r.
DOI: https://doi.org/10.7554/eLife.28620.040
**Source data 1.** RNA-seq MA plot of Olfr Expression in mOSNs versus Fusion Protein expressing mOSNs.txt.
DOI: https://doi.org/10.7554/eLife.28620.041
**Source data 2.** RNA-seq Log2 fod change in mOSNs versus Fusion Protein Expressing mOSNs.txt.
DOI: https://doi.org/10.7554/eLife.28620.042
**Source data 3.** Change in ATAC-seq signal in Fusion Protein expressing mOSNs by peak type.txt.
DOI: https://doi.org/10.7554/eLife.28620.043
**Source data 4.** Change in OR expression in Fusion Protein expressing mOSNs versus promoter motifs.txt.
DOI: https://doi.org/10.7554/eLife.28620.044
**Figure supplement 1.** Genetic strategy for expression of Fusion Protein.
DOI: https://doi.org/10.7554/eLife.28620.036
**Figure supplement 2.** Fusion protein expression most strongly affects ATAC-seq signal on Greek Islands.
DOI: https://doi.org/10.7554/eLife.28620.037
**Figure supplement 3.** Fusion protein expression downregulates OR expression irrespective of presence of Lhx2 or Ebf promoter motifs.
DOI: https://doi.org/10.7554/eLife.28620.038

There is also a uniform reduction of OR expression independently of the distance between the OR and the closest Greek Island, and this reduction occurs irrespective of the motif content of OR promoters (*Figure 6C,D*). Moreover, comparable downregulation was observed for the ORs with a Greek Island in the promoter region (distance = 1) and for ORs that lack a Greek Island in cis (distance set to 1e + 08) (*Figure 6C,D*). Thus, functional incapacitation of Greek Islands by two distinct genetic manipulations results in specific but pervasive disruption of OR expression irrespective of OR promoter sequence, OR distance from a Greek Island, presence of a Greek Island within the OR cluster, or even presence of a Greek island within the same chromosome.

If *trans* interactions between Greek Islands are essential for OR transcription and the formation of a multi-enhancer hub over a stochastically chosen OR allele is the low probability event responsible for singular OR choice, then increasing the number of Greek Islands in an OR cluster should increase the expression frequency of the ORs in that cluster. To test this prediction, we introduced, by homologous recombination, an array of 5 Greek Islands (*Lipsi, Sfaktiria, Crete, H and Rhodes*, hereafter termed *LSCHR*) next to the endogenous *Rhodes*, a Greek Island from chromosome 1 (*Figure 7A*). This array comprised the ATAC-seq accessible core of each Greek Island (392–497 bp) together with 50 bp of endogenous flanking sequence (*Supplementary file 6*). We chose Rhodes for this manipulation for two reasons: First, the ATAC-seq and ChIP-seq signals on *Rhodes* are among the strongest between the 63 Greek Islands, which combined with the almost complete H3K9me3 local depletion suggest that it is accessible and bound by Lhx2 and Ebf in the majority of mOSNs. Thus, any transcriptional changes observed by this manipulation would not be attributed to increased Lhx2 and Ebf binding on this locus. Second, there are no additional Greek Islands within a genomic distance of over 80 MB on chromosome 1, thus formation of a Greek Island hub over this cluster requires recruitment of unlinked OR enhancers. We, therefore, reasoned that Rhodes-proximal ORs would be

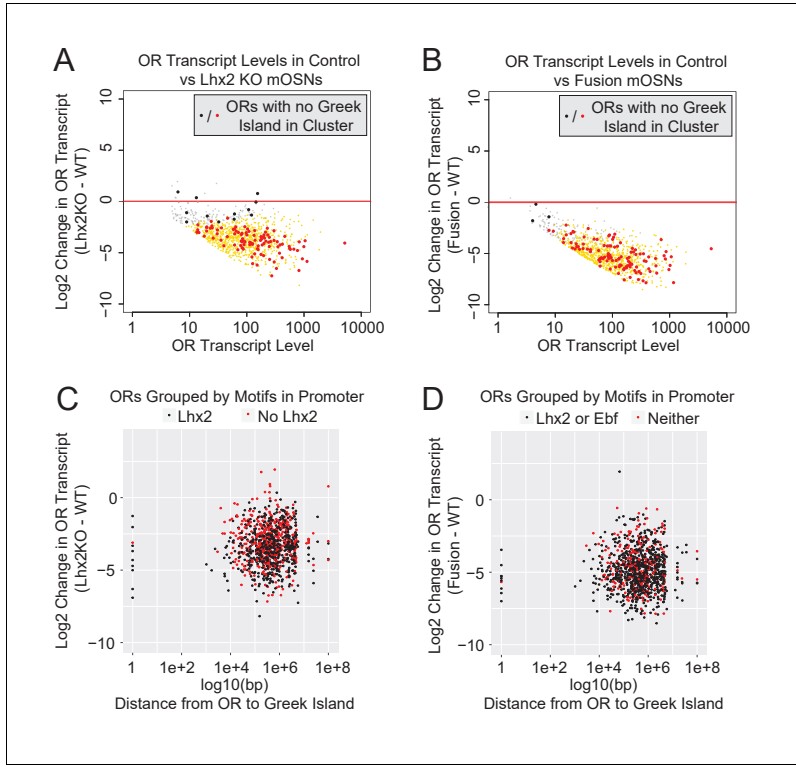

**Figure 6.** Downregulation of OR expression over large genomic distances. (**A**) MA-plot of OR transcript levels in FAC-sorted *Lhx2* KO (*Omp-IRES-Cre; Lhx2fl/fl; tdTomato*) mOSNs compared to FAC-sorted control mOSNs (*Omp-IRES-GFP*). Gold dots correspond to OR genes with statistical significant transcriptional changes. ORs in clusters without a Greek Island are shown as large dots, with significantly changed ORs in red. Three biological replicates were included for control mOSNs and 2 biological replicates were included for *Lhx2* KO mOSNs. (**B**) MA-plot of OR transcript levels in FAC-sorted Fusion protein expressing (*Omp-IRES-tTA; tetO-Fusion-2a-mcherry*) mOSNs compared to FAC-sorted control mOSNs (*Omp-IRES-GFP*). Gold dots correspond to OR genes with statistical significant transcriptional changes. ORs in clusters without a Greek Island are shown as large dots, with significantly changed ORs in red. Three biological replicates were included for control mOSNs and data from 2 independent founders were included for the Fusion Protein transgene. See *Figure 6—figure supplement 1* for an example OR cluster without a Greek Island. (**C**) Plot of OR distance from a Greek Island compared to Log2 Fold change in *Lhx2* KO mOSNs. ORs overlapping a Greek Island have distance set to 1. ORs on a chromosome without a Greek Island have distance set to 1e + 08. (**D**) Plot of OR distance from a Greek Island compared to Log2 Fold change in Fusion Protein expressing mOSNs. ORs overlapping a Greek Island have distance set to 1. ORs on a chromosome without a Greek Island have distance set to 1e + 08.

DOI: https://doi.org/10.7554/eLife.28620.045

The following source data, source code and figure supplements are available for figure 7:

**Source code 1.** R code for analysis of RNA-seq data from Lhx2KO mOSNs.r.
DOI: https://doi.org/10.7554/eLife.28620.047

**Source code 2.** R code for analysis of RNA-seq data from Fusion Protein mOSNs.r.
DOI: https://doi.org/10.7554/eLife.28620.048

**Source data 1.** RNA-seq MA-plot of OR expression in Lhx2KO versus presence of Greek Island.txt.
DOI: https://doi.org/10.7554/eLife.28620.049

**Source data 2.** RNA-seq MA-plot of OR expression in Fusion Protein versus presence of Greek Island.txt.
DOI: https://doi.org/10.7554/eLife.28620.050

**Source data 3.** OR expression in Lhx2KO versus promoter motifs and distance to Greek Island.txt.
DOI: https://doi.org/10.7554/eLife.28620.051

**Source data 4.** OR expression in Fusion Protein mOSNs versus promoter motifs and distance to Greek Island.txt.
DOI: https://doi.org/10.7554/eLife.28620.052

**Figure supplement 1.** Fusion Protein and *Lhx2* KO downregulate ORs in a cluster without a Greek Island.
DOI: https://doi.org/10.7554/eLife.28620.046

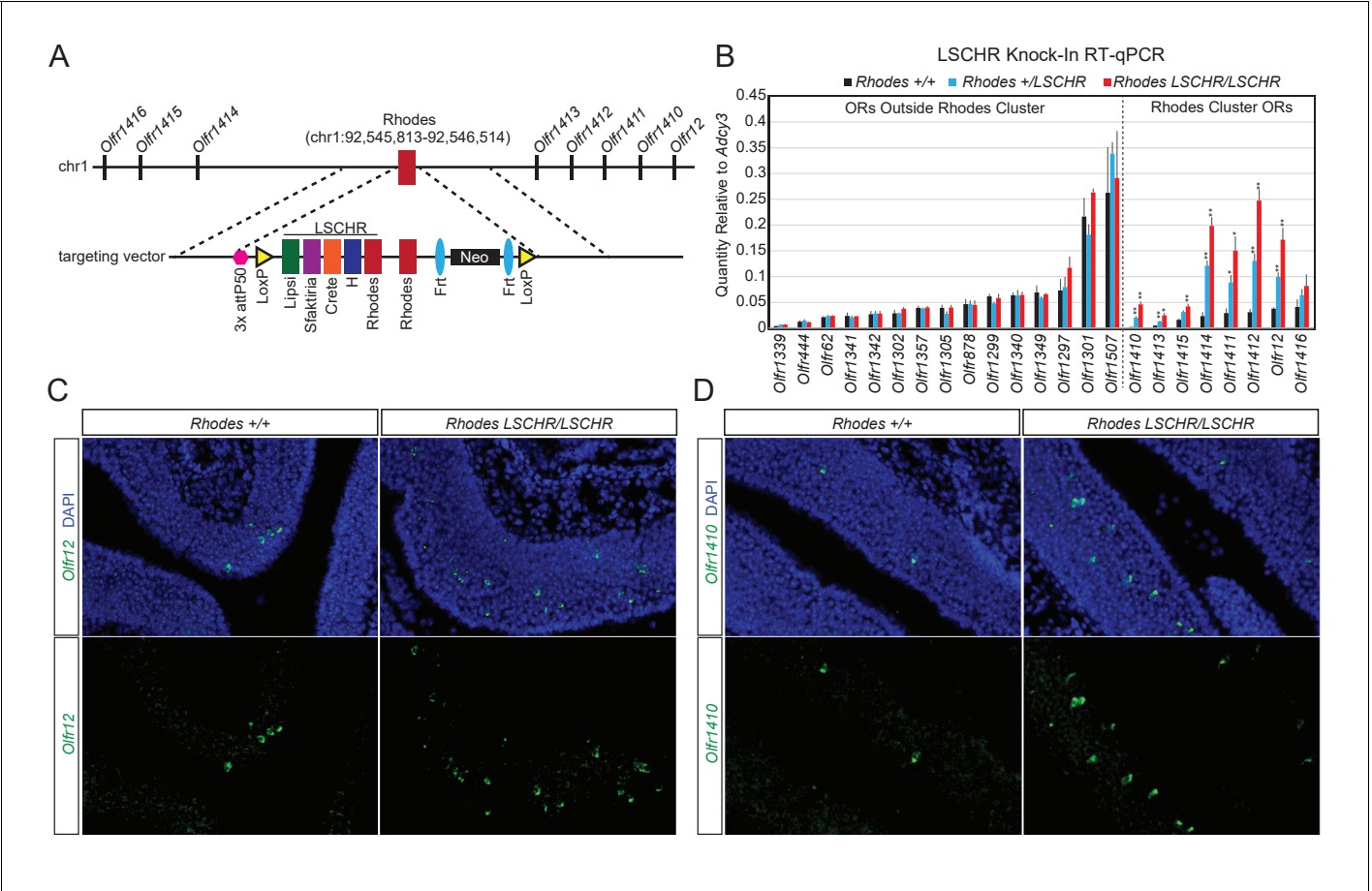

**Figure 7.** Multi-enhancer hubs activate OR transcription. (A) Targeted insertion of 5 Greek Islands (*LSCHR*) adjacent to *Rhodes*. Coordinates are mm10. See *Figure 7—figure supplement 1* for ChIP qPCR analysis of Lhx2 binding to the inserted Greek Islands. (B) RT-qPCR of OR transcript levels in MOEs of 3 week old LSCHR mice and wild-type littermate controls. Transcript levels are expressed as quantity relative to *Adcy3*, error bars are SEM. ORs are grouped by presence inside or outside the OR cluster containing *Rhodes*, and within each group ORs are ordered by level of expression in wild-type mice. *p<0.05, **p<0.01, two-tailed student's t-test. For wild-type mice n = 3, for LSCHR heterozygous and homozygous mice n = 4. (C) Fluorescent RNA in situ hybridization with probe for *Olfr12* (green) in *LSCHR* homozygous and wild-type littermate control MOE at 2 weeks of age. Nuclei are labeled with DAPI (blue). (D) Fluorescent RNA in situ hybridization with probe for *Olfr1410* (green) in *LSCHR* homozygous and wild-type littermate control MOE at 2 weeks of age. Nuclei are labeled with DAPI (blue).

DOI: https://doi.org/10.7554/eLife.28620.053

The following figure supplement is available for figure 7:

**Figure supplement 1.** Lhx2 binds to inserted Greek Islands.
DOI: https://doi.org/10.7554/eLife.28620.054

more responsive to the insertion of additional enhancers next to their local Greek Island, than ORs residing on chromosomes with multiple Greek Islands.

To test whether transcription factors bind the Greek Islands comprising the *LSCHR* knock-in allele in vivo we performed ChIP-qPCR for Lhx2 from whole MOE of mice homozygous for the *LSCHR* knock-in and wild-type controls. When normalized to percent input, which adjusts for the increased copy number of each Greek Island, we observe strong binding of Lhx2 to the *LSCHR* Greek Islands that is similar to the binding observed in wild-type mice (*Figure 7—figure supplement 1A*). If a reference set of external control sites are instead used for normalization, we detect an approximately two-fold increase in *LSCHR* Greek Island signal in the input control and in the Lhx2 ChIP (*Figure 7—figure supplement 1B,C*), consistent with the increase of the number of alleles of the 5 Greek Islands in the knock-in mice. Taken together, these data suggest that insertion of a Greek Island array next to Rhodes does not increase significantly the frequency of Lhx2 binding to Rhodes or to the transgenic Greek Island alleles.

Despite the minimal effects of *LSCHR* insertion on transcription factor binding, we detect significant transcriptional upregulation of the OR alleles in the Rhodes cluster. q-PCR analysis of cDNA prepared from the whole MOE of LSCHR knock-in mice and wild type littermates, shows strong transcriptional upregulation of the ORs in the Rhodes cluster that is almost doubled in homozygote knock-in mice in comparison to heterozygote littermates (*Figure 7B*). ORs from different clusters and non-OR genes are not strongly upregulated by this manipulation; however, four of the ORs in the *Rhodes* cluster are upregulated by more than 8 fold in the homozygote knock-in mice (*Figure 7B*). In fact, *Olfr1412*, which is the most upregulated OR in the Rhodes cluster approaches mRNA levels comparable to *Olfr1507*, the most highly expressed OR in the MOE (*Figure 7B*). RNA FISH experiments demonstrate that this transcriptional upregulation represents an increase in frequency of choice, rather than an increase of transcription rates in each cell (*Figure 7B,D*). ORs from different clusters do not appear significantly affected by this genetic manipulation, a result that is not surprising since the *trans* effects of this enhancer array would be distributed to more than a 1000 OR genes.

## Discussion

In most cell types interchromosomal interactions are rare and thus far appear to represent technical or biological noise (*Nagano et al., 2015*), rather than to provide a reliable mechanism for gene regulation. Various studies suggest that the majority of genomic interactions are restricted within topologically associated domains (TADs) that show little variation between different tissues (*Dixon et al., 2012*). Specific genomic interactions between TADs are infrequent, and interactions between different chromosomes are even less prominent (*Lieberman-Aiden et al., 2009*; *Rao et al., 2014*). However, in certain biological contexts, specific interchromosomal interactions are readily detected by imaging and genomic approaches, or have been inferred genetically. For example, during X chromosome inactivation, there is a 'chromosome kissing' step that occurs just before one of the two chromosomes is inactivated (*Bacher et al., 2006*; *Masui et al., 2011*; *Xu et al., 2006*). During T and B cell differentiation interchromosomal interactions regulate antigen receptor choice and cellular differentiation (*Hewitt et al., 2008*; *Spilianakis et al., 2005*). The stochastic induction of the human *IFN beta* gene by virus infection requires the formation of interchromosomal interactions between the *IFN beta* enhancer and NF-kappa B-bound Alu repeats (*Apostolou and Thanos, 2008*). Finally, stochastic photoreceptor choice in drosophila omatidia is determined by DNA elements that, genetically, appear to communicate in trans (*Johnston and Desplan, 2014*). Thus, although interchromosomal interactions may not be involved in gene regulation in most cell types, their stochastic and infrequent nature may be ideal for the execution of non-deterministic, and mutually exclusive regulatory processes like OR gene choice (*Dekker and Mirny, 2016*).

The involvement of interchromosomal interactions in OR gene choice was first postulated by the demonstration that the prototypical OR enhancer, the *H* enhancer (*Serizawa et al., 2003*), interacts in trans with transcriptionally active ORs (*Lomvardas et al., 2006*). The significance of these interactions was questioned as deletion of the H enhancer affected the expression of only three proximal ORs (*Fuss et al., 2007*; *Nishizumi et al., 2007*). Subsequently, however, additional OR enhancers, the Greek Islands, were discovered to a current total of 63 elements. The striking similarities between these elements in regards of the transcription factors that bind to them, combined with the demonstration that Greek Islands form a complex network of interchromosomal interactions (*Markenscoff-Papadimitriou et al., 2014*), suggested that extensive functional redundancy may mask the effects of single or even double (*Khan et al., 2011*) enhancer deletions in trans. The non-redundant role of Greek Islands for the expression of certain ORs in cis may be attributed to the inability of some OR promoters to recruit enhancers from other chromosomes, making them completely dependent on the presence of a proximal enhancer for this function. In other words, even if *trans* enhancement is required for the activation of every OR gene, a fraction of them may depend on the assistance of a local Island for the recruitment of *trans* enhancers. Such qualitative promoter differences are consistent with the observation that enhancer deletions affect only some ORs in a cluster, and by the fact that certain ORs can be expressed as transgenic minigenes (*Vassalli et al., 2002*), while others can be expressed as transgenes only in the presence of an enhancer in cis (*Serizawa et al., 2003*). The proposed redundant function of Greek Islands as *trans* enhancers may have facilitated the rapid evolution of this gene family, which expanded dramatically during the

transition from aquatic to terrestrial life (*Niimura and Nei, 2007*). Activation of OR transcription by Greek Islands in trans allows the functional expression of newly evolved OR alleles in mOSNs, without a requirement for physical linkage to an enhancer- a property fully compatible with gene expansion through retrotransposition, segmental duplication, and chromosomal translocation. Thus, OR gene activation through non-deterministic genomic interactions in trans may provide a mechanism that 'shuffles the deck' and assures that a newly evolved OR allele will be expressed at a frequency similar to that of the existing OR repertoire.

## Global and *trans* action of OR enhancers

A correlation between the formation of interchromosomal Greek Island hubs and OR transcription was previously established by ectopic expression of Lamin b receptor (Lbr) in mOSNs, and by conditional deletion of transcriptional co-activator Bptf, either of which caused reduction of Greek Island interactions in trans and pervasive OR downregulation (*Clowney et al., 2012*; *Markenscoff-Papadimitriou et al., 2014*). However, these manipulations have more general consequences that extend beyond the regulation of Greek Island interaction. For example, ectopic Lbr expression in mOSNs caused a general rearrangement of nuclear topology and disrupted the aggregation of OR clusters, making difficult to distinguish between the effects on interchromosomal OR clustering and interchromosomal Greek Island interactions. Deletion of *Bptf* on the other hand, although it only disrupted interchromosomal associations between Greek Islands, also caused a developmental arrest in the OSNs that may, or may not, be related to the failure to activate OR expression.

To minimize indirect effects that may confound the interpretation of these manipulations, we targeted a common and highly specific genetic signature among Greek Islands, the composite motif. This DNA sequence constitutes a remarkable example of highly constrained and stereotypically distributed transcription factor binding motifs that is shared between most Greek Islands, and is highly enriched relative to OR promoters and co-bound sites genome-wide. Overexpression of a 'synthetic' fusion protein that specifically recognizes the composite motif eliminated ATAC-seq signal from Greek Islands in mOSNs, suggesting that it displaced the endogenous Lhx2 and Ebf proteins on most OR enhancers. Similar observations were made for the conditional Lhx2 deletion, which also reduced the chromatin accessibility of Greek Islands and abolished Ebf binding from these elements. The strong and specific downregulation of the OR transcriptome in both *Lhx2* knock out and in fusion protein expressing mOSNs, clearly reveals the critical and ubiquitous role of the Greek Islands as key regulators of OR expression. The fact that these transcriptional effects extend to ORs that have neither a Greek Island in cis nor Lhx2/Ebf motifs on their promoters, is consistent with the role of Greek Islands as *trans* OR gene enhancers. However, it should be noted that we cannot exclude alternative interpretations of our data. For example, it is possible that there are additional intergenic OR enhancers with *bona fide* composite sites that are only utilized in neurons that express specific OR genes. Although our analysis in the 3 purified OSN populations expressing ORs Olfr1507, Olfr17 and Olfr171 does not support the existence of OR-specific Greek Islands, we cannot exclude the possibility that Greek Islands with such restricted activity exist in other OSN sub-populations. Furthermore, although our ATAC-seq analysis suggests that fusion protein overexpression affects predominantly Greek Islands, indirect effects on OR transcription cannot be excluded. Taking all these caveats into account, the fact that distinct genetic manipulations that target the Greek Islands, and their genomic interactions, cause widespread downregulation of OR expression, provides strong genetic support for the requirement of interchromosomal interactions in OR gene choice.

## Same transcription factors different chromatin states

The experimental demonstration that every Greek Island is co-bound by Lhx2 and Ebf, the same transcription factors predicted to bind on most OR promoters, is unexpected because of the fundamentally different chromatin states of the two types of regulatory elements in mOSNs. OR promoters are inaccessible in the mixed mOSN population, and only upon FAC-sorting cells that express the same OR, could we obtain evidence for OR promoter accessibility. In contrast, the enhancers of OR genes appear accessible and bound by Lhx2 and Ebf in a large fraction of mOSNs. The stereotypically proximal positioning of Lxh2 and Ebf motifs on OR enhancers emerged as the key determinant for these differences, since the functionally cooperative binding of Lhx2 and Ebf on proximal motifs in vivo appears to counteract the propagating properties of the surrounding

heterochromatin. Notably, cooperative binding between 'phased' Lhx2 motifs was recently proposed as an explanation for the increased frequency of expression of OR transgenes under the control of an artificial promoter(*D'Hulst et al., 2016*), although in this case the local chromatin state is not known. Interestingly, the composite Lhx2/Ebf motif that we identified on Greek Islands is structurally very similar to the numerous heterodimeric motifs identified by an in vitro screen for sequences that are co-bound by a variety of transcription factors (*Jolma et al., 2015*). Thus, the solution that was adopted by intergenic OR enhancers to generate heterochromatin-resistant binding sites, may be generally utilized by other transcription factors in a variety of genomic contexts and regulatory needs. In support of this, the striking stereotypy of Lhx2 and Ebf motifs in Greek Islands, also known as 'rigid motif grammar' (*Long et al., 2016*), is reminiscent of the constraint spacing of transcription factor binding sites in the *IFN beta* enhanceosome (*Panne et al., 2007*; *Thanos and Maniatis, 1995*).

## A multi-enhancer hub for robust and singular OR expression

The concept that Greek Islands may have stronger affinity for Lhx2 and Ebf than OR promoters, immediately provides a molecular solution for the need of a multi-enhancer hub for stable and robust OR transcription. In the event that an OR promoter becomes de-silenced and occupied by Lhx2 and Ebf, singular or weak binding by these transcription factors will be unstable, due to the competing forces of flanking heterochromatin. However, if an OR promoter is surrounded by multiple strong sites of cooperative binding, like the ones we detect in high frequency on the Greek Islands, then every time Lhx2 and Ebf fall off an OR promoter they will be sequestered by local, high affinity sites, which may also act as a replenishing source for these transcription factors. In other words, inter-chromosomal Greek island hubs may create local regions of high Lhx2 and Ebf concentration that is essential for continuous binding on the low affinity sites of a chosen OR promoter and high transcription rates.

Thus, we propose a model whereby the deployment of multiple, individually weak components that function in a coordinated and hierarchical fashion to activate OR transcription. According to this model, first, cooperative interactions between Lhx2 and Ebf result in stable binding to Greek Islands, which prevents flanking heterochromatin from spreading and silencing these intergenic elements. Because composite motifs are specifically enriched on Greek Islands similar cooperative interactions between Lhx2 and Ebf cannot protect OR promoters from heterochromatic silencing (*Figure 8A*). Second, cooperative interactions between Greek Islands assemble numerous Lhx2 and Ebf elements into a multi-chromosomal enhancer hub (*Figure 8B*). When this hub forms stable interactions with a stochastically chosen OR allele in trans, then heterochromatin is

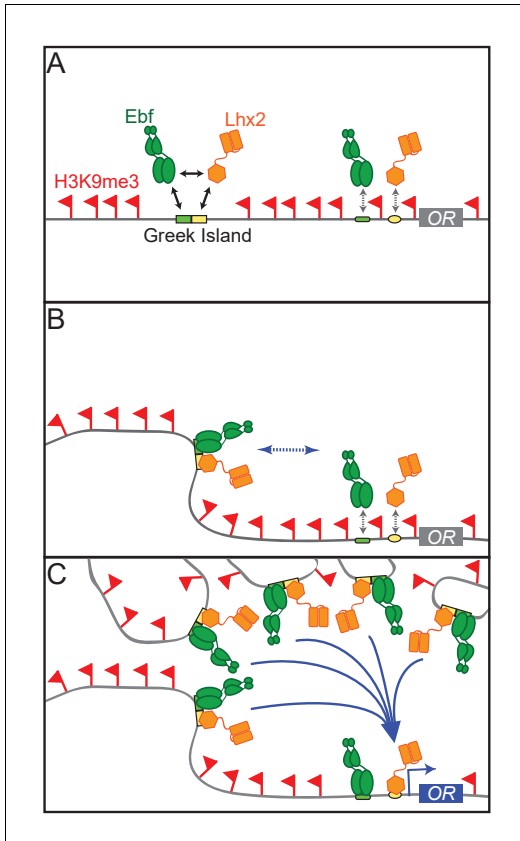

**Figure 8.** A Hierarchical Model for OR gene choice. (**A**) Lhx2 and Ebf bind in a functionally cooperative fashion on the composite motifs of the Greek Islands. Because these motifs are not juxtaposed in most OR promoters, Lhx2 and Ebf cannot overcome the heterochromatic silencing of OR promoters, thus their binding is restricted to the OR enhancers. (**B**) Lhx2/Ebf bound OR enhancers are not strong enough to activate proximal OR alleles on their own and to facilitate stable transcription factor binding on their promoters. (**C**) Lhx2/Ebf bound Greek Islands form an interchromosomal, multi-enhancer hub that recruits coactivators essential for the de-silencing of OR promoters and robust transcriptional activation of the OR allele that would be recruited to this hub.
DOI: https://doi.org/10.7554/eLife.28620.055

displaced, and cooperative enhancer-promoter interactions mediate stable Lhx2 and Ebf binding on the promoter, and therefore, transcriptional activation (*Figure 8C*). These cooperative interactions may be direct, homotypic interactions between Lhx2 and Ebf or facilitated by coactivator or mediator proteins that are recruited by these transcription factors. In either scenario, the same fundamental principles of cooperativity and synergy that govern the genetic switch between lysis and lysogeny in the lambda bacteriophage (*Ptashne, 2009*), and promote the formation and function of the human *IFN beta* enhanceosome (*Thanos and Maniatis, 1995*), may also regulate the formation of a 3-dimensional enhanceosome responsible for OR gene choice.

A multi-enhancer hub model explains why the few OR promoters that are bound by Lhx2 and Ebf in a large fraction of mOSNs are not transcribed at higher frequencies than most ORs. It also may explain why 63 OR genes, one for each Greek Island, are not simultaneously expressed in each mOSN: if numerous enhancers must cooperate for OR transcription, individual promoters, and even individual enhancer-promoter combinations, may not be sufficient for OR transcription. But what prevents the formation of numerous multi-enhancer hubs, which could then activate more than one OR allele at a time? The answer to this critical question may be found in the transcriptional phenotype of the Rhodes knock-in mice, whereby 6 Greek Islands reside in tandem. In these mice, we detect a significant increase in the frequency of OR choice, that is not caused by increased occupancy by Lhx2. Thus, the increase frequency of local OR choice is likely explained by a mechanism subsequent to transcription factor binding; either through the more efficient recruitment of an unknown limited coactivator, or by recruitment of additional *trans* enhancers, culminating at the assembly of more complex, transcription-competent Greek island hub. Regardless of the mechanism by which *LSCHR* increases the frequency of OR choice within the *Rhodes* cluster, the local ORs remain silent in the vast majority of the mOSNs. This implies the existence of a strong 'thresholding' mechanism in the ability of Greek Islands to activate OR transcription, such that even 6 Islands acting together are inadequate to drive ubiquitous expression in most mOSNs. Thus, even if multiple enhancer hubs were to form in an mOSN nucleus, only the ones that surpass a critical number of interacting Greek Islands would lead to the activation of OR transcription. These non-linear properties are reminiscent of 'super-enhancers' (SE), which exhibit steep concentration threshold requirements attributed to phase transition processes(*Hnisz et al., 2017*). Thus, an attractive model suggests that Greek Island hubs may undergo phase transition when they exceed a steep threshold of interacting Greek Islands. If this phase transition is required for OR transcription, hubs with lower complexity will not be transcriptionally competent, providing an elegant molecular solution of the singularity of OR gene choice.

Such thresholding mechanism may be less strict in immature OSNs and progenitors, where low level OR co-expression is detected by single cell RNA-seq (*Hanchate et al., 2015*; *Saraiva et al., 2015*; *Tan et al., 2015*). Similar low level co-expression is detected in Lbr-expressing mOSNs, where the nuclear aggregation of OR clusters is prevented and the chromatin accessibility of OR genes is increased (*Clowney et al., 2012*). Thus, it is possible that the differentiation dependent silencing and aggregation of heterochromatic OR clusters into condensed nuclear foci contribute to this 'all or none' transcriptional paradigm. In other words, the extreme silencing forces imposed by mOSNs on OR genes may result in extraordinary requirements for OR transcription, which can only be met by an activating multi-enhancer assembly of unprecedented complexity and possibly unique biochemical properties. Thus, even if more than one multi-enhancer hub could form in a nucleus, the number of transcription-competent hubs would be extremely limited if not singular. Combined with the kinetic restrictions imposed by the OR-elicited feedback signal, and a recently reported post choice refinement process(*Abdus-Saboor et al., 2016*), our model provides a mechanistic solution for the singular choice of one out of >2000 OR alleles.

## Materials and methods

### Contact for reagent and resource sharing

Further information and requests for resources and reagents should be directed to and will be fulfilled by the Lead Contact, Stavros Lomvardas (sl682@columbia.edu)

## Experimental model and subject details

### Mice

Mice were treated in compliance with the rules and regulations of IACUC under protocol number AC-AAAI1108. All experiments were performed on dissected whole main olfactory epithelium (MOE) or on freshly isolated, FAC sorted primary cells collected from whole main olfactory epithelium.

Mature olfactory sensory neurons (mOSNs) were sorted from *Omp-IRES-GFP* mice, which were previously described (*Shykind et al., 2004*). Olfr17+ cells were sorted from *Olfr17-IRES-GFP* mice (*Shykind et al., 2004*). Olfr151+ cells were sorted from *Olfr151-IRES-tauGFP* mice (Olfr151tm26Mom) (*Bozza et al., 2002*). Olfr1507+ cells were sorted from *Olfr1507-IRES-GFP* mice (Olfr1507tm2Rax) (*Shykind et al., 2004*).

Conditional deletion of Lhx2 in mOSNs was achieved by crossing *Lhx2* conditional allele mice (*Lhx2*fl/fl: Lhx2tm1.1Monu) (*Mangale et al., 2008*) with *Omp-IRES-Cre* mice (Omptm1(cre)Jae) (*Eggan et al., 2004*). In order to sort *Lhx2* knock out mOSNs, a Cre-inducible tdtomato allele (Gt (ROSA)26Sortm14(CAG-tdTomato)Hze/J ) (*Madisen et al., 2010*) was also included in this cross.

Transgenic mice bearing the *TetO-Fusion-2A-mCherry* construct were generated at the Columbia University Transgenic Mouse facility at the Irving Cancer Research Center. Fusion protein expression in mOSNs was achieved by crossing these mice with *Omp-IRES-tTA* mice (Omptm1(tTA)Gogo)(*Yu et al., 2004*).

Rhodes knock-in mice were generated by Biocytogen.

For ATAC-seq and ChIP-seq experiments, cells were sorted from adult male and female mice ranging in age from 7 to 16 weeks of age. For RNA-seq, cells were sorted from male and female mice ranging in age from 6 to 10 weeks.

Biological replicate samples were collected and processed separately from different mice.

## Method details

### Fluorescence activated cell sorting

Mice were sacrificed using $CO_2$ followed by cervical dislocation. The main olfactory epithelium (MOE) was dissected and transferred to ice-cold EBSS (Worthington Biochemical). The MOE was cut in to small pieces with a razor blade, and then dissociated with a papain dissociation system (Worthington Biochemical). Diced tissue was added to papain-EBSS, with at most 2 MOEs/mL, and incubated for 40 min at 37°C on a rocking platform. After 40 min, tissue was triturated 30 times, the supernatant containing dissociated cells was transferred to a new tube, and the cells were pelleted (300 rcf, 5 min, room temperature). Remaining papain was inhibited by resuspending the cell pellet with Ovomucoid protease inhibitor solution diluted 1:10 in EBSS, and the dissociated cells were pelleted (300 rcf, 5 min, room temperature).

For live cell sorts, dissociated cells were washed once with sort media (PBS with 2% Fetal Bovine Serum), and then resuspended in sort media supplemented with 100 U/mL DNase I (Worthington Biochemical), 4 mM MgCl₂, and 500 ng/mL DAPI (Invitrogen). These cells were passed through a 40 uM cell strainer, and then FAC sorted. Live cells were selected by gating out DAPI positive cells.

For formaldehyde-fixed cell sorts, dissociated cells were resuspended in PBS + 1% methanol-free formaldehyde (Pierce). Cells were fixed at room temperature for 5 min, and then fixation was quenched by adding 1/10th volume of 1.25M glycine. Fixed cells were pelleted (500 rcf, 5 min, room temperature), washed once with sort media, resuspended in sort media, passed through a 40 uM cell strainer, and then FAC sorted.

### Preparation of cross-linked chromatin

Sorted fixed cells were pelleted (800 rcf,10 minutes, 4°C). Cell pellets were resuspended in ChIP Lysis Buffer (50 mM Tris-HCl pH 7.5, 150 nM NaCl, 0.5% NP-40, 0.25% Sodium Deoxycholate, 0.1% SDS, 1x protease inhibitors (Sigma, 05056489001)) and incubated on ice for 10 min. Nuclei were collected by centrifugation (1000 rcf, 5 min, 4°C). The nuclei pellet was resuspended in shearing buffer (10 mM Tris-HCl pH 7.5, 1 mM EDTA, 0.25% SDS, 1x protease inhibitors) and then sheared to a size range of 200–500 bp on a Covaris S220 Focused-ultrasonicator (16 min, 2% Duty Cycle, Peak Power 105W, 200 cycles per burst, 6°C). Sheared chromatin was centrifuged (10,000 rcf, 10 min, 4°C) to remove insoluble material. The DNA concentration of the sheared chromatin was determined by

fluorescent quantification (ThermoFisher, P7589). Shearing was assessed by agarose gel electrophoresis after DNA clean-up: chromatin was incubated for 30 min with RNase A, cross-links were reversed overnight at 65 C, and then DNA was column purified (Zymo Research, D4014).

## Chromatin immunoprecipitation of cross-linked chromatin

Formaldehyde cross-linked chromatin was used for ChIP of Ebf (Aviva, ARP32960_P050, RRID: AB_2045685) and Lhx2 (*Roberson et al., 2001*). Approximately 2 ug of sheared chromatin was diluted to 500 uL with ChIP Buffer (16.7 mM Tris-HCl pH 8.1, 167 mM NaCl, 1.2 mM EDTA, 1.1% Triton X-100, 0.01% SDS, 1x protease inhibitors), and then pre-cleared with Protein G Dynabeads (Life Technologies) for one hour at 4°C. After preclearing, the supernatant containing cleared chromatin was transferred to a new tube, and approximately 100 ng of chromatin was set aside as an input control. Input control chromatin was stored at 4°C until the elution step. The remaining pre-cleared chromatin was incubated overnight at 4°C with 1 ug of Ebf antibody or 1 uL of Lhx2 antibody. Protein G beads were blocked overnight with 2 mg/ml yeast tRNA (Life Technologies) in ChIP Buffer. The next day, the blocked beads were washed once with ChIP Buffer, then resuspended in antibody bound chromatin. Chromatin was incubated with beads for 1–2 hr at 4°C with rotation. Chromatin bound beads were washed 5 times with LiCl Wash Buffer (100 mM Tris-HCl pH 7.5, 500 mM LiCl, 1% NP-40, 1% Sodium Deoxycholate) and once with TE pH 7.5. DNA was eluted from beads by incubating at 65°C for 30 min with 100 uL ChIP Elution Buffer (1% SDS, 0.1M NaHCO$_3$ 4 mM DTT) in a thermomixer set to 900 rpm. This elution was repeated and the elution fractions were pooled. The eluted DNA was incubated overnight at 65°C. Input chromatin was brought up to 200 uL with elution buffer and also incubated at 65°C overnight. ChIP DNA and input DNA were column purified using Zymo ChIP DNA columns (Zymo Research, D5205) and eluted in 20 uL of 10 mM Tris-HCl pH 8. Purified DNA was analyzed by quantitative PCR (see *Supplementary file 8* for primer sequences) and used to generated sequencing libraries.

## Micrococcal nuclease digestion

Live sorted cells were pelleted (800 rcf, 15 min, 4°C) and then resuspended in Buffer 1 (0.3M Sucrose, 60 mM KCl, 15 mM NaCl, 5 mM MgCl$_2$, 0.1 mM EGTA, 15 mM Tris-HCl pH 7.5, 5 mM Sodium Butyrate, 0.1 mM PMSF, 0.5 mM DTT, 1x protease inhibitors). Cells were lysed by adding an equal volume of Buffer 2 (0.4% Igepal CA-630, 0.3M Sucrose, 60 mM KCl, 15 mM NaCl, 5 mM MgCl$_2$, 0.1 mM EGTA, 15 mM Tris-HCl pH 7.5, 5 mM Sodium Butyrate, 0.1 mM PMSF, 0.5 mM DTT, 1x Protease Inhibitor Cocktail). After addition of Buffer 2, cells were incubated on ice for 10 min, and then nuclei were pelleted (1000 rcf, 10 min, 4°C). Nuclei were resuspended in MNase buffer (0.32M Sucrose, 4 mM MgCl$_2$, 1 mM CaCl$_2$, 50 mM Tris-HCl pH 7.5, 5 mM Sodium Butyrate, 0.1 mM PMSF, 1x protease inhibitors). Nuclei were digested for 1 min and 40 s with 0.2U of Micrococcal Nuclease (Sigma) per 1 million cells. Digestion was stopped by adding EDTA to a final concentration of 20 mM. Undigested material was pelleted (10,000 rcf, 10 min, 4°C), and the supernatant (S1 fraction) was retained and stored at 4°C. The pelleted material was resuspended in Dialysis Buffer (1 mM Tris-HCl pH 7.5, 0.2 mM EDTA, 5 mM Sodium Butyrate, 0.1 mM PMSF, 1x protease inhibitors), rotated overnight at 4°C. Following dialysis, the insoluble material was pelleted (10,000 rcf, 10 min, 4°C) and the supernatant (S2 fraction) was retained. MNase digestion was assessed by agarose gel electrophoresis. The MNase treatment was optimized to yield a nucleosomal ladder comprising mostly mono and di-nucleosome sized fragments in the S1 fraction and di-nucleosome and larger sized fragments in the S2 fraction. The concentration of nucleic acid in the S1 and S2 fractions was determined by fluorescent quantification (ThermoFisher, P7589). Prior to ChIP equal volumes of S1 and S2 fractions were combined, and the total quantity of nucleic acid in the pooled fractions calculated to normalize between experiments.

## Native chromatin immunoprecipitation

MNase digested native chromatin was used for ChIP with antibodies for H3K9me3 (Abcam, ab8898, RRID:AB_306848), H3K79me3 (Abcam ab2621, RRID:AB_303215), and H3K27ac (Active Motif, AM39133). Approximately 1 ug of MNase digested chromatin was used per IP, with approximately 100 ng reserved as an input control. Chromatin was diluted to 500 uL in Wash Buffer 1 (50mM Tris-HCl pH 7.5, 10 mM EDTA, 125 mM NaCl, 0.1% Tween-20, 5 mM Sodium Butyrate, 1x protease

inhibitors), 1 ug of antibody was added, and the binding reaction was rotated overnight at 4°C. For each IP, 10 uL of Protein A Dynabeads (Life Technologies) and 10 uL of Protein G Dynabeads were blocked overnight with 2 mg/ml yeast tRNA and 2 mg/mL BSA in Wash Buffer 1. Blocked beads were added to antibody bound chromatin and rotated for 1–2 hr at 4°C. Bound beads were washed 4 times with Wash Buffer 1, 3 times with Wash Buffer 2 (50mM Tris-HCl pH 7.5, 10 mM EDTA, 175 mM NaCl, 0.1% Igepal CA-630, 5 mM Sodium Butyrate, 1x protease inhibitors), and once with TE pH7.5. DNA was eluted from beads by incubating at 37°C for 15 min with 100 uL Native ChIP Elution Buffer (10 mM Tris-HCl pH7.5, 1 mM EDTA, 1% SDS, 0.1 M NaHCO$_3$) in a thermomixer set to 900 rpm. This elution was repeated and the elution fractions were pooled. Input chromatin was brought up to 200 uL with Native ChIP Elution Buffer. ChIP DNA and input DNA were column purified using Zymo ChIP DNA columns (Zymo Research, D5205) and eluted in 20 uL of 10 mM Tris-HCl pH 8.

## ChIP-seq library preparation
ChIP-seq libraries were prepared with Nugen Ovation Ultralow v2 kits.

## ATAC-seq
ATAC-seq libraries were prepared from live sorted cells using the protocol developed by *Buenrostro et al., 2015*. Cells were pelleted (500 rcf, 5 min, 4°C) and then resuspended in lysis buffer (10 mM Tris-HCl, pH 7.4, 10 mM NaCl, 3 mM MgCl2, 0.1% IGEPAL CA-630). Nuclei were immediately pelleted (1000 rcf, 10 min, 4°C). Pelleted nuclei were resuspended in transposition reaction mix prepared from Illumina Nextera reagents (for 50 uL: 22.5 uL water, 25 uL 2xTD buffer, 2.5 uL Tn5 Transposase). The volume of the Tn5 transposition reaction was scaled to the number of cells collected: 1 uL mix per 1000 cells. If fewer than 10,000 cells were collected by FACS, 10 uL scale reactions were performed. See *Supplementary file 4* for a summary of ATAC-seq experiments. Transposed DNA was column purified using a Qiagen MinElute PCR cleanup kit (Qiagen). The transposed DNA was then amplified using barcoded primers and NEBNext High Fidelity 2x PCR Master Mix (NEB). Amplified libraries were purified using Ampure XP beads (Beckman Coulter) at a ratio of 1.6 ul of beads per 1 uL of library and eluted in 30 uL of elution buffer (10 mM Tris-HCl pH 8, 0.1 mM EDTA).

## qRT-PCR
MOEs from 3 week old mice were dissected, cut in to small pieces with a razor blade, and then added to 1 mL of Trizol. Samples were vortexed for 15 s, and then incubated for 5 min at room temperature. Total RNA was extracted by adding 200 uL chloroform, vortexing for 15 s, incubating at room temperature for 2 min, then centrifugation at 12,000 rcf for 15 min at 4°C. The aqueous phase was collected and RNA was precipitated with isopropyl alcohol with 10 ug/mL linear acrylamide (ThermoFisher) added as a carrier. The RNA pellet was washed twice with 75% ethanol, dried, then resuspended in nuclease free water. 3 ug of RNA was DNase treated using the TURBO DNA-free Kit (ThermoFisher) according to manufacturer's instructions. cDNA was prepared from 800 ng of RNA using SuperScriptIII (ThermoFisher) and used for qPCR with gene specific primers (*Supplementary file 8*). Fold change was calculated using the ddCT approach, using Adcy3 as a reference gene to normalize between samples and expressing fold change relative to a wild type littermate control.

## RNA-seq
Live sorted cells were pelleted (15 min, 800 rcf, 4°C), the supernatant was aspirated until 250 uL of media remained, and then the cell pellet was resuspended in 750 uL Trizol LS (ThermoFisher). Total RNA was extracted by adding 200 uL chloroform, vortexing for 15 s, incubating at room temperature for 2 min, then centrifugation at 12,000 rcf for 15 min at 4°C. The aqueous phase was collected and RNA was precipitated with isopropyl alcohol with 10 ug/mL linear acrylamide (ThermoFisher) added as a carrier. The RNA pellet was washed twice with 75% ethanol, dried, then resuspended in nuclease free water. 1 ug of RNA was DNase treated using the TURBO DNA-free Kit (ThermoFisher) according to manufacturer's instructions. RNA-seq libraries were prepared from DNase-treated RNA using a TruSeq Stranded Total RNA with Ribo-Zero Gold Set B kit (Illumina RS-1222302).

## Deep sequencing

Sequencing libraries were profiled on Bioanalyzer 2100 using a high sensitivity DNA kit (Agilent). Library concentration was determined by KAPA assay (KAPA Biosystems). Libraries were multiplexed and sequenced on an Illumina HiSeq with 50 bp single-end or paired-end reads. See *Supplementary file 5* for a summary of sequencing data.

## Recombinant DNA

Three versions of the fusion protein were designed with either 1, 2, or 4 repeats of a 5 amino acid linker sequence between the DNA binding domain of Ebf and the DNA binding domain of Lhx2 (*Supplementary file 6*). Gene blocks encoding these proteins were synthesized by Integrated DNA Technologies. Gene blocks were TOPO cloned into a pcDNA3.1/V5-His expression vector (Thermo-Fisher). For in vivo expression, the fusion protein was subcloned into a pTRE2 vector that was modified to include a sequence encoding t2A-mCherry.

The 5-enhancer hub that was inserted into the Rhodes locus was generated using Gibson assembly of gene blocks synthesized by Integrated DNA Technologies (*Supplementary file 6*).

## Electrophoretic mobility shift assay

Probe oligonucleotides (*Supplementary file 7*) were annealed, gel purified, and end labeled with $^{32}$P using T4 Polynucleotide kinase. Labeled probes were purified on a Sephadex G-50 column (GE Healthcare 27-5330-01).

Fusion protein, Lhx2, and Ebf were in vitro translated from pcDNA3 expression vectors bearing the T7 promoter (Promega, L1170). In vitro binding reactions were setup with 1 uL of in vitro translation product, 0.5 ug Poly(dI-dC), 1 ug BSA, and 1xProtease Inhibitor cocktail in EMSA Binding Buffer (10 mM HEPES pH 7.5, 40 mM KCl, 5% Glycerol, 0.5% Igepal CA-630, 1 mM DTT). For competition conditions, 2 pmol (20-fold excess) or 10 pmol (100-fold excess) of unlabeled, annealed oligonucleotides was added to the binding reactions. The binding reactions were incubated for 20 min on ice, and then 100 fmol of radiolabeled probe was added. Following probe addition, binding reactions were incubated at room temperature for 10 min. Binding reactions were loaded on a native TBE polyacrylamide gel (6% acrylamide, 49:1 bis-acrylamide:acrylamide), and electrophoresed at 180 V for approximately 3 hr. After running, gels were transferred to filter paper, dried, and exposed to a phosphoimager screen overnight. Phosphoimager screens were scanned on a Typhoon FLA7000 or FLA9500.

## RNA in situ hybridization

Regions of Olfr12 and Olfr1410 were cloned and verified by Sanger sequencing (*Supplementary file 6*). DNA for in vitro transcription was generated by PCR from these templates using an antisense primer bearing the T7 promoter. RNA probe was generated by in vitro transcription of 1 ug of PCR product with T7 polymerase and Fluorescein RNA Labeling Mix (Sigma). Probe RNA was ethanol precipitated and resuspended in 50 uL of hybridization buffer (50% formamide, 5x SSC, 5x Denhart's, 250 ug/mL Yeast tRNA, 500 ug/mL Salmon Sperm DNA). Prior to hybridization, probe was diluted 40x in hybridization buffer and denatured at 85 C for 5 min.

For RNA Fluorescent In Situ Hybridization (*Figure 7C,D*), MOE was dissected, embedded in OCT (ThermoFisher), and then frozen. Coronal cryosections were taken at a thickness of 10 um and then air dried for 10 min. Slides were fixed with 4% PFA in PBS for 10 min. After fixation, slides were rinsed three times with PBS, and then washed with PBST (PBS with 0.1% Triton X-100) for 10 min. Slides were then rinsed once with PBS and then incubated for 15 min in Acetylation Buffer (0.021 N HCl, 1.2% Triethanolamine (v/v), 0.25% Acetic Anhydride (v/v)). After acetylation, slides were rinsed 3 times with PBS, then probe was added and hybridized overnight at 65 C in a humid chamber. Following hybridization, slides were washed twice for 15 min with 0.2% SSC at 65 C, rinsed three times with PBS, and then blocked for 1 hr with TNB (0.1M Tris pH 7.5, 0.15M NaCl, 0.05% Blocking Reagent (Perkin Elmer)). After blocking, slides were stained overnight at 4 C with anti-fluorescein POD antibody (Roche) diluted 1:100 in TNB. The next day, slides were rinsed twice with TNT buffer (0.1M Tris pH7.5, 0.15M NaCl, 0.1% Tween 20) and then washed in TNT buffer for 30 min. Slides were then treated with TSA amplification with Fluorescein labeling for 4 min, and then washed 6

times for 5–10 min with TNT buffer. DAPI was included in the final TNT wash at a concentration of 2.5 ug/mL. Slides were then mounted with Vectashield (Vector Laboratories) and imaged.

## Immunofluorescence

For imaging GFP and mCherry, MOE was dissected and fixed in 4% PFA for 30 min on ice prior to being embedded in OCT. Coronal cryosections were taken at a thickness of 12 to 14 um and then air dried for 10 min. Slides were fixed with 4% PFA for 10 min. After fixation, slides were washed with PBST (PBS with 0.1% Triton X-100), blocked in PBST-DS (PBST +4% donkey serum), stained with DAPI (2.5 ug/mL) in PBST-DS, washed with PBST, and then mounted with Vectashield and imaged.

For immunofluorescence (*Figure 4A*), MOE was dissected from 3 week old mice, embedded in OCT (ThermoFisher), and then frozen. Cryosections were taken and slides were fixed and washed as described above. Slides were stained with primary antibody (a-Lhx2, diluted 1:1000) in PBST-DS overnight at 4°C. Slides were then washed, stained with DAPI (2.5 ug/mL) and secondary antibody (donkey a-rabbit conjugated to Alexa-488, diluted 1:1000, ThermoFisher) in PBST-DS for 1 hr, washed, and then mounted with Vectashield and imaged.

## Microscopy

Confocal images were collected with a Zeiss LSM 700. Image processing was carried out with ImageJ (NIH).

## Quantification and statistical analysis

### ChIP-seq and ATAC-seq sequencing data processing and analysis

Adapter sequences were removed from raw sequencing data with CutAdapt (RRID:SCR_011841). ChIP-seq and ATAC-seq reads were aligned to the mouse genome (mm10) using Bowtie2 (RRID: SCR_006646) (*Langmead and Salzberg, 2012*). Default settings were used, except a maximum insert size of 1000 (-X 1000) was allowed for ATAC-seq and native ChIP-seq data since these data sets contained some large fragments. PCR duplicate reads were identified with Picard (RRID:SCR_006525) and removed with Samtools (RRID:SCR_002105) (*Li et al., 2009*). Samtools was used to select uniquely aligning reads by removing reads with alignment quality alignments below 30 (-q 30).

For Lhx2 and Ebf ChIP-seq, HOMER (RRID:SCR_010881) (*Heinz et al., 2010*) was used to call peaks of ChIP-seq signal using the 'factor' mode and an input control. Consensus peak sets were generated by selecting peaks that overlapped between biological replicates and extending them to their combined size. For signal tracks, replicate experiments were merged, and HOMER was used to generate 1 bp resolution signal tracks normalized to a library size of 10,000,000 reads.

For H3K9me3, H3K79me3, and H3K27ac ChIP-seq replicate experiments were merged, and HOMER was used to generate 1 bp resolution signal tracks normalized to a library size of 10,000,000 reads. Regions enriched for H3K9me3 were identified by running HOMER peak calling in region mode, with the following settings: -L 0 -F 2.5 -size 2000 -minDist 4000. A consensus set of H3K9me3 enriched regions was generated by selecting regions that were enriched in both biological replicates.

For ATAC-seq, regions of open chromatin were identified by running HOMER peak calling in 'region' mode, with a fragment size of 150 bp and a peak size of 300 bp. For ATAC-seq signal tracks, replicate experiments were merged, and HOMER was used to generate 1 bp resolution signal tracks normalized to a library size of 10,000,000 reads. Reads were shifted 4 bp upstream in order to more accurately map the Tn5 insertion site. Reads were extended to the full fragment length, as determined by paired-end sequencing, except for signal tracks of ATAC-seq fragment ends (*Figure 3C*), which were generated by using a fragment size of 1 bp.

A small number of failed ChIP-seq and ATAC-seq experiments were excluded from analysis. Failed experiments were identified based upon the presence of very few enriched peaks and low fold enrichment of reads in the identified peaks.

### Classification of Greek Islands

The set of Greek Islands with function in Zebrafish or Mouse transgene assays (*Markenscoff-Papadimitriou et al., 2014*) was examined using data from *Omp-IRES-GFP* sorted mOSNs. With the

exception of P, all were bound by Ebf and Lhx2, and were present within a region enriched for H3K9me3. Using these criteria, we used Bedtools2 (*Quinlan and Hall, 2010*) to generate a new, comprehensive list of sites with these properties in mOSNs. Specifically, the Greek Islands are defined as sites with overlapping peaks of Ebf and Lhx2 binding, within a region enriched for H3K9me3, and within an OR cluster (*Supplementary file 1*).

## OR annotation

Annotation of OR gene transcripts was take from *Ibarra-Soria et al. (2014)*. ORs absent from this annotation but present in the UCSC mm10 annotation were added. For OR gene heatmaps, transcripts were merged by OR gene and the most 5' annotated TSS and most 3' annotate TES were used.

## Lhx2 and Ebf co-binding

The background rate of overlap between Ebf and Lhx2 peaks was calculated for the whole genome excluding peaks within H3K9me3 positive regions of OR clusters. For Ebf, 4729 peaks overlapped Lhx2 peaks, whereas 4230 did not. In H3K9me3 positive regions of OR clusters, 63 out of 65 Ebf peaks overlapped Lhx2 peaks. In R, this overlap rate (63 out of 65), was compared to the genome-wide rate (4729/8850) using a Binomial test, with the alternative hypothesis that the overlap rate is greater in OR clusters, yielding a p-value of p=2.557e-16. For Lhx2, 4729 peaks overlapped Ebf peaks, whereas 11468 did not. In H3K9me3 positive regions of OR clusters, 63 out of 114 Lhx2 peaks overlapped Ebf peaks. In R, this overlap rate (63 out of 114), was compared to the genome-wide rate (4729/16197) using a Binomial test, with the alternative hypothesis that the overlap rate is greater in OR clusters, yielding a p-value of p=5.702e-09.

## Heatmaps and signal plots

Heatmaps and signal plots were generated with Deeptools2(*Ramírez et al., 2016*). Unless otherwise specified, heatmaps were sorted by mean signal over the interval shown.

## GO analysis

GREAT (*McLean et al., 2010*) was used for gene ontology analysis of sites bound by Lhx2 and Ebf in mOSN.

## Motif analysis

Motif analysis was performed with HOMER. Motif searches were run on Ebf and Lhx2 consensus peak sets for the 200 bp region around the center of the peak, with repeat masking. The top de novo identified motif for Lhx2 and Ebf ChIP-seq was converted to TRANSFAC format, and plotted using Weblogo v3.5 (*Crooks et al., 2004*), together with motifs derived from published Ebf (GEO: GSE21978, *Lin et al., 2010*) and Lhx2 (GSE48068, [*Folgueras et al., 2013*]) ChIP-seq data.

HOMER was used to search Greek Island sequences for motifs with a variety of settings. A search for long motifs that allowed up to 4 mismatches (-len 18,20 -mis 4) recovered a sequence motif that was highly enriched relative to random background sequences (p=1e-59). HOMER was then used to optimize this motif against a background set of all mOSN Ebf and Lhx2 co-bound sites. This optimized motif is reported as the composite motif in *Figure 3A*. The HOMER derived composite motif was converted to TRANSFAC format and plotted using Weblogo v3.5.

For *Figure 3B*, HOMER was used to analyze sets of sequences and identify the highest scoring match to the composite motif in each sequence. The cumulative distribution of scores for each set were plotted in R. The binomial distribution was used to calculate the statistical significance the enrichment of composite motif sequence in Greek Island sequences relative to Ebf and Lhx2 Co-bound sites genome-wide.

To analyze specific instances of the composite motif for *Figure 3C–F*, HOMER was used to identify all instances of the composite motif scoring above a given threshold within Greek Islands. For *Figure 3D and E*, the DNA sequence of Greek Island composite motifs, together with 20 bp of sequence on either side, was converted to a matrix and visualized with deeptools2.

Composite motif multiple alignments (*Figure 3—figure supplement 1A,B*) were generated with Jalview (*Waterhouse et al., 2009*).

## Motif proximity analysis

HOMER was used to identify all instances of the de novo, ChIP-seq derived Ebf and Lhx2 motifs genome-wide. Bedtools was then used to identify all instances of Ebf motifs that occur within Greek Islands with a composite motif score of 10 or above. Greek Islands without an Ebf motif were excluded from further analysis. For the remaining islands, Bedtools2 was used to identify the closest Lhx2 motif to each of Ebf motif. For Greek Islands with multiple Ebf motifs, only the closest pair was retained. Additional sets of sequences were analyzed in the same manner, and the distribution of motif distances was plotted in R. An identical analysis centered on Lhx2 motifs rather than Ebf motifs yielded similar results.

## Quantitative analysis of normalized ChIP-seq and ATAC-seq data

Normalized ATAC-seq and Ebf ChIP-seq data were generated in R using the Diffbind package (RRID:SCR_012918) (*Ross-Innes et al., 2012*). Diffbind was used to generate a read count for each peak for each data set. Count data was normalized using the 'DBA_SCORE_TMM_READS_EFFEC-TIVE' scoring system, which normalizes using edgeR and the effective library size. After normalization, counts for biological replicates were averaged, and then counts were log transformed for plotting. For analyses comparing ChIP-seq data from control mOSNs, which were sequenced with paired-ends, to data from *Lhx2*KO mOSNs, which were sequenced from a single end, the raw data from control mOSN was re-aligned using only read 1.

## RNA-seq data processing and analysis

Adapter sequences were removed from raw sequencing data with CutAdapt. RNA-seq reads were aligned to the mouse genome (mm10) using STAR (*Dobin et al., 2013*). Samtools was used to select uniquely aligning reads by remove reads with alignment quality alignments below 30 (-q 30). Signal tracks were generated with RSeQC (RRID:SCR_005275) (*Wang et al., 2012*), either retaining strand information (-d '+-,-+') (*Figure 4—figure supplement 1*), or without strand information (*Figure 5F* and *Figure 6—figure supplement 1*). RNA-seq signal plots are normalized to a library size of 1,000,000 reads. The Sashimi plot in *Figure 4—figure supplement 1* was generated using IGV (*Robinson et al., 2011*).

RNA-seq data analysis was performed in R with the DESeq2 package (*Love et al., 2014*). Genes with no counts in any condition were excluded. For *Figure 1—figure supplement 5*, DEseq2 was used to calculate FPKM values, and these values were plotted for subsets of OR genes. For all other plots, differential gene expression analysis was run comparing control mOSNs and *Lhx2*KO or Fusion protein expressing mOSNs. The base mean and log2fold change values from these analyses were used for plots. For MA-plots, significantly changed genes were identified with an adjusted p-value cutoff of 0.05.

For density and scatter plots of log2 fold change in OR transcript levels (*Figure 4—figure supplement 3*, *Figure 5—figure supplement 3*, *Figure 6C*, *Figure 6D*), ORs with low levels of expression (Normalized Base Mean <5) were excluded.

## Data and software availability

### Data resources

The data reported in this paper will be available through GEO with accession number GSE93570.

## Acknowledgements

We would like to thank Drs. Richard Axel, Tom Maniatis, Richard Mann, and members of the Lomvardas lab for input, suggestions, and discussions and for critical reading of the manuscript. We are grateful to Dr. Tom Jessell for the extensive use of his FAC-sorting instrument, to Dr. Mark Roberson for the anti-Lhx2 antibody and to Dr. Abbas Rizvi for assistance with statistical calculation. KM was funded by F32 post-doctoral fellowship GM108474 (NIH). This project was funded by R01DC013560 and R01DC015451, R01 DA036894 (NIH), and the HHMI Faculty Scholar Award.

## Additional information

### Funding

| Funder | Grant reference number | Author |
|---|---|---|
| National Institute of General Medical Sciences | GM108474 | Kevin Monahan |
| National Institute on Deafness and Other Communication Disorders | | Stavros Lomvardas |
| Howard Hughes Medical Institute | | Stavros Lomvardas |

The funders had no role in study design, data collection and interpretation, or the decision to submit the work for publication.

### Author contributions

Kevin Monahan, Conceptualization, Supervision, Funding acquisition, Writing—original draft, Project administration, Writing—review and editing; Ira Schieren, Performed FACS for every experiment in the manuscript; Jonah Cheung, Designed of the fusion protein based on analysis of crystallographic data; Alice Mumbey-Wafula, Mouse genotyping, set up of genetic crosses, histological preparations; Edwin S Monuki, Generated and provided the Lhx2 fl/fl mouse strains; Stavros Lomvardas, Resources, Methodology, Optimized and performed all the FACS experiments

### Author ORCIDs

Kevin Monahan, http://orcid.org/0000-0001-8922-5801
Stavros Lomvardas, http://orcid.org/0000-0002-7668-3026

### Ethics

Animal experimentation: This study was performed in strict accordance with the recommendations in the Guide for the Care and Use of Laboratory Animals of the National Institutes of Health. All of the animals were handled according to approved institutional animal care and use committee (IACUC) protocols AC-AAAI1108 of Columbia University.

### Decision letter and Author response

Decision letter https://doi.org/10.7554/eLife.28620.067
Author response https://doi.org/10.7554/eLife.28620.068

## Additional files

### Supplementary files

• Supplementary file 1. New versus Old Greek Islands.xlsx
DOI: https://doi.org/10.7554/eLife.28620.056

• Supplementary file 2. Greek Island Composite Motifs.xlsx
DOI: https://doi.org/10.7554/eLife.28620.057

• Supplementary file 3. Olfactory Recpetor Clusters.xlsx
DOI: https://doi.org/10.7554/eLife.28620.058

• Supplementary file 4. ATAC-seq Samples.xlsx
DOI: https://doi.org/10.7554/eLife.28620.059

• Supplementary file 5. Deep Sequencing Samples.xlsx
DOI: https://doi.org/10.7554/eLife.28620.060

• Supplementary file 6. Recombinant DNA.xlsx
DOI: https://doi.org/10.7554/eLife.28620.061

• Supplementary file 7. EMSA Probe Sequences.xlsx
DOI: https://doi.org/10.7554/eLife.28620.062

• Supplementary file 8. Primer Sequences.xlsx
DOI: https://doi.org/10.7554/eLife.28620.063
• Transparent reporting form
DOI: https://doi.org/10.7554/eLife.28620.064

## Major datasets

The following dataset was generated:

| Author(s) | Year | Dataset title | Dataset URL | Database, license, and accessibility information |
|---|---|---|---|---|
| Monahan K, Lomvardas S | 2017 | Chromatin State and Binding of Lhx2 and Ebf in Olfactory Sensory Neurons | https://www.ncbi.nlm.nih.gov/geo/query/acc.cgi?acc=GSE93570 | Publicly available at the NCBI Gene Expression Omnibus (accession no: GSE93570) |

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
