## [Decision Letter]

Thank you for submitting your article "Cooperative interactions enable singular olfactory receptor expression" for consideration by *eLife*. Your article has been reviewed by two peer reviewers, and the evaluation has been overseen by a Reviewing Editor and a Senior Editor. The following individuals involved in review of your submission have agreed to reveal their identity: Yin Shen (Reviewer #3).

The reviewers have discussed the reviews with one another and the Reviewing Editor has drafted this decision to help you prepare a revised submission.

Summary:

In this study, Lomvardas and colleagues carried out a series of experiments to test the hypothesis that cooperative binding of the Lhx2 and Ebf transcription factors to OR enhancer elements is required for the singular OR gene expression in olfactory sensory neurons. The authors present compelling data demonstrating the enrichment of contiguous Lhx2 and Ebf binding motifs in the "Greek Island" elements linked to OR gene loci and their association with open chromatin as assayed by ATAC-seq; and likely cooperative binding of these two transcription factors to the combined motif as evidenced by the diminished binding of Ebf's in the Lhx2 conditional knockout. To demonstrate necessity of the proposed cooperative interaction, the authors then devised an ingenious experiment in which they first designed a linked Ebf-Lhx2 fusion protein that shows preferential binding to the combined motif and then use it to block OR expression through its transgenic expression in neurons in vivo. Sufficiency was inferred by observing an increase in expression of locus-specific ORs in a knock-in mouse in which a tandem repeat of Greek Island elements was inserted into the OR locus containing Rhodes.

This work provides genetic insight for the stochastic mono-allelic OR gene expression program, and offer support to the model of inter-chromosomal interactions via the Greek Island elements in OR gene regulation. In addition, the results presented here suggest that cooperative binding of Lhx2 and Ebf transcription factors is involved in OR gene choice, an aspect of OR gene regulation that thus far has remained somewhat elusive by comparison to data supporting mechanisms of negative feedback regulation and stability of OR expression.

Minor points:

Reviewers raised the following points for further improvement of the manuscript:

a) The Greek Island insertion experiment shown in Figure 6 showed an increase in expression of linked ORs. In order to make this a stronger case, it would be nice to show that the ectopic Greek Islands also exhibit open chromatin by ATAC-seq and Lhx2 and Ebf binding by ChIP-seq.

b) The authors ought to be more open-minded when interpreting their results in the context of their model in which the Greek Islands mediate inter-chromosomal interactions required for high level, singular OR expression. It remains to be determined whether and how Lhx2-Ebf binding plays a role in such complexes. Indeed, if anything the data of Figure 6 clearly show an effect of the Greek Island array in cis but not in trans. It is a not an unreasonable leap to infer meaning to trans interactions, but again a more circumspect discussion would seem more appropriate. Discussion of alternative models would be important.

c) In Figure 5, the authors show expression of Lhx2 DBD and Ebf DBD fusion protein affect the chromatin state at Greek island and affect OR gene expression. Though non-OR gene expression is not affected globally, it will be nice to show whether such fusion protein could affect the chromatin states outside of the Greek islands.

d) In Figure 6, KI of an array Greek islands increases the frequency of OR gene activation (Rhodes OR genes). This suggests that having the insertion of Greek island is not sufficient for gene activation in all cells. What is the percentage of neurons that exhibit OR gene activation? Could they authors discuss why such insertion can only achieve gene activation in a subset of the mature olfactory neurons? What are the other possible players for controlling OR gene activation?

---

## [Author Response]

Minor points:Reviewers raised the following points for further improvement of the manuscript:a) The Greek Island insertion experiment shown in Figure 6 showed an increase in expression of linked ORs. In order to make this a stronger case, it would be nice to show that the ectopic Greek Islands also exhibit open chromatin by ATAC-seq and Lhx2 and Ebf binding by ChIP-seq.

In our design of this knock-in experiment we did not introduce heterologous sequences that would allow us to distinguish between the endogenous Greek Islands from the ones inserted in the Rhodes cluster. Thus, we have no way to differentiate the state of the ectopic and endogenous enhancers in these mice. However, we reasoned that if the tandem insertion of 5 Greek Islands next to Rhodes increases occupancy by transcription factors we would see an in increase in the ChIP enrichment. ChIP-qPCR comparing Lhx2 occupancy on the 5 Greek Islands in wild type and homozygote knock-in mice does not show significant changes in Lhx2 enrichment on the 5 Islands between the two genotypes (except of a small increase of Lhx2 binding on Lipsi). These results, together with various normalization methods that take into account the existence of extra enhancer alleles, are shown in Figure 7—figure supplement 1. These data are consistent with our initial prediction that the insertion of additional enhancer elements will not increase the accessibility and transcription factor occupancy of a locus that is already accessible in most olfactory neurons. Thus, the increased frequency of choice of proximal OR alleles is most likely due to effects subsequent to transcription factor binding (either more efficient recruitment of a limited co-activator or increased interactions with other Greek Islands in trans, both of which are discussed in the text).

b) The authors ought to be more open-minded when interpreting their results in the context of their model in which the Greek Islands mediate inter-chromosomal interactions required for high level, singular OR expression. It remains to be determined whether and how Lhx2-Ebf binding plays a role in such complexes. Indeed, if anything the data of Figure 6 clearly show an effect of the Greek Island array in cis but not in trans. It is a not an unreasonable leap to infer meaning to trans interactions, but again a more circumspect discussion would seem more appropriate. Discussion of alternative models would be important.

Changes are made in the Discussion and alternative explanations and possible caveats of our data are provided.

c) In Figure 5, the authors show expression of Lhx2 DBD and Ebf DBD fusion protein affect the chromatin state at Greek island and affect OR gene expression. Though non-OR gene expression is not affected globally, it will be nice to show whether such fusion protein could affect the chromatin states outside of the Greek islands.

We added a new panel showing that the strongest effects in chromatin accessibility are observed on Greek Islands, consistent with the transcriptional changes following expression of the fusion protein (Figure 5—figure supplement 2).

d) In Figure 6, KI of an array Greek islands increases the frequency of OR gene activation (Rhodes OR genes). This suggests that having the insertion of Greek island is not sufficient for gene activation in all cells. What is the percentage of neurons that exhibit OR gene activation? Could they authors discuss why such insertion can only achieve gene activation in a subset of the mature olfactory neurons? What are the other possible players for controlling OR gene activation?

As shown in the in situ hybridization experiments there are very few neurons in each section that express ORs from the Rhodes cluster, both in wild type and mutant mice. Thus, the main point of this experiment is that in each section of the olfactory epithelium the majority of the OSNs do not express the ORs from the Rhodes cluster. In other words, insertion of 5 strong transcriptional enhancers next to Rhodes does not result in the transcriptional dominance of the neighboring ORs. Determining the exact frequency of olfactory neurons expressing these ORs in an accurate fashion is beyond the point, since in both wild type and mutant mice they are expressed in less than 1% of the total population.

Finally, we split the original Figure 6 into two figures: Figure 6 for the computational analysis of the effects of Lhx2 deletion and fusion protein overexpression; Figure 7 for the description of the 5 Island insertion next to Rhodes.